# PERICLIMv1.0: A model deriving palaeo-air temperatures from thaw depth in past permafrost regions

Tomáš Uxa[1], Marek Křížek[2], and Filip Hrbáček[3]

[1]Institute of Geophysics, Czech Academy of Sciences, Praha, Czech Republic
[2]Department of Physical Geography and Geoecology, Faculty of Science, Charles University, Praha, Czech Republic
[3]Department of Geography, Faculty of Science, Masaryk University, Brno, Czech Republic

**Correspondence:** Tomáš Uxa (uxa@ig.cas.cz)

**Abstract.** Periglacial features, such as various kinds of patterned ground, cryoturbations, frost wedges, solifluction structures, or blockfields, are among the most common relics of cold-climate periods, which repetitively occurred throughout the Quaternary, and, as such, they are widespread archives of past environmental conditions. Climate controls on the development of most periglacial features, however, remain poorly known, and thus empirical palaeo-climate reconstructions based on them have a limited validity. This study presents and evaluates a new simple inverse modelling scheme PERICLIMv1.0 (PERIglacial CLIMate) that derives palaeo-air temperature characteristics related to the palaeo-active-layer thickness, which can be recognized using many relict periglacial features found in past permafrost regions. The evaluation against modern temperature records showed that the model reproduces air temperature characteristics with average errors $\leq 1.3\,^\circ$C. Besides, the past mean annual air temperature modelled experimentally for two sites in the Czech Republic hosting relict cryoturbation structures was between $-7.0\pm1.9\,^\circ$C and $-3.2\pm1.5\,^\circ$C, which is well in line with earlier reconstructions utilizing various palaeo-archives. These initial results are promising and suggest that the model could become a useful tool for reconstructing Quaternary palaeo-environments across vast areas of mid- and low-latitudes where relict periglacial assemblages frequently occur, but their full potential remains to be exploited.

## 1 Introduction

Many mid- and low-latitude regions of the world host a number of distinctive landforms and subsurface structures collectively termed as relict periglacial features, such as various kinds of patterned ground, cryoturbations, frost wedges, solifluction structures, or blockfields, which developed during cold periods of the Quaternary due to intense freeze-thaw activity taking place under seasonal-frost or permafrost conditions. Commonly, these features occur in places where other palaeo-indicators are rare or absent, or emerged at different times, which enhances their relevance as archives of past environmental conditions (Washburn, 1980; French, 2017; Ballantyne, 2018). So far, relict periglacial features have been used to reconstruct past climates almost exclusively on the basis of climate thresholds of their active counterparts that are mostly situated in present-day high-latitude periglacial environments (Ballantyne and Harris, 1994). Such empirical interpretations are, however, problematic because suitable analogues for past periglacial environments are rare particularly due to substantial contrasts in solar insolation between mid- or low- and high-latitudes (Williams, 1975; French, 2017), and even if they can be found, active periglacial fea-

tures that occur there may have developed under different climate conditions than those prevailing at the present time (Uxa et al., 2017; Ballantyne, 2018). Climate controls on the development of most periglacial features are thus poorly known, usually implying broad ranges of climate conditions (Washburn, 1980; Harris, 1982, 1994; Karte, 1983; Wayne, 1983; Ballantyne and Harris, 1994; Huijzer and Isarin, 1997; Ballantyne, 2018), which also relates to the fact that the features partly depend on other factors, such as ground physical properties, hydrology, topography, or ground-surface cover (Ballantyne, 2018). Conse-

quently, the inferred palaeo-climates have frequently been thought to be of limited validity, and indeed most periglacial features have been widely accepted only as indicators of seasonal frost or permafrost and ground-ice presence (Ballantyne and Harris, 1994; Ballantyne, 2018). This situation can be largely attributed to prevailing interest in mapping the distribution patterns of periglacial features and their associations with mean annual air temperature (MAAT), pervading traditional palaeo-periglacial geomorphology, while other details on their surface and subsurface dimensions have been widely overlooked. Greater empha-

sis on the surface and subsurface attributes of periglacial features, which are closely related to their formation and responsible processes, could, however, advance the discipline far beyond its current frontiers (cf. Barsch, 1993; French and Thorn, 2006).

Periglacial features form through various thermally- and gravity-induced processes that mostly operate within a layer of seasonal freezing and thawing, the base of which is commonly sharply defined and confines the subsurface dimensions of the features (Williams, 1961). This zone is thus usually discernible in vertical cross-sections because intense ice segregation and

mass displacements associated with the formation of periglacial features alter the freeze-thaw layer so that its composition and properties differ from those of the underlying ground. This contrast may be preserved long after the periglacial features have ceased to be active and, as such, it can indicate the thickness of the palaeo-freeze-thaw layer (French, 2017). Since the freeze-thaw depth closely couples with ground and air temperature conditions (e.g., Frauenfeld et al., 2004; Åkerman and Johansson, 2008; Wu and Zhang, 2010), it retains a valuable palaeo-climate record that can be approximated based on modern

air temperature–freeze-thaw depth relations (Williams, 1975) or can be retrieved through an inverse solution of the equations calculating the freeze-thaw depth (Maarleveld, 1976; French, 2008). Obviously, this idea is not new, but despite its ingenuity, simplicity, and general acceptance in a benchmark periglacial literature (e.g., Washburn, 1979; Ballantyne and Harris, 1994; French, 2017; Ballantyne, 2018), it has never been developed into a viable tool for deriving past thermal regimes that has a sound mathematical basis, is replicable, and lacks subjectivity (see Williams, 1975; Maarleveld, 1976), because computational

methods have been durably underused by periglacial geomorphologists interested in reconstructions of Quaternary palaeo-environments.

This study presents and evaluates a simple modelling scheme PERICLIMv1.0 (PERIglacial CLIMate) that is designed to infer palaeo-air temperature characteristics associated with relict periglacial features indicative of the palaeo-active-layer thickness, and discusses its uncertainties and applicability with respect to other palaeo-proxy records and/or model products.

It specifically targets on palaeo-active-layer phenomena because their palaeo-environmental significance as well as preservation potential is substantially higher than for seasonal-frost features. Besides, it intends to stimulate the application of modelling tools and foster the development of new quantitative methods in palaeo-environmental reconstructions utilizing relict periglacial features in order to raise their reputation as palaeo-proxy indicators.

## 2 Model description

The PERICLIMv1.0 builds on an inverse solution of the Stefan (1891) equation, which has originally been developed to determine the thickness of sea ice, but it also well describes the thaw propagation in ice-bearing grounds, and lately it has become probably the most commonly used analytical tool for estimating the thickness of the active layer over permafrost (e.g., Klene et al., 2001; Shiklomanov and Nelson, 2002; Hrbáček and Uxa, 2020). It assumes that the thawed-zone temperature, which is controlled by the ground-surface temperature at the surface boundary, decreases linearly towards the bottom frozen

zone that is constantly at 0 °C and latent heat is the only energy sink associated with its thawing, that is, heat conduction below the thaw front is not accounted for (Kurylyk, 2015). As such, the Stefan equation tends to deviate inversely proportional to the moisture content in the active layer as well as the active-layer temperature at the onset of thawing (Romanovsky and Osterkamp, 1997; Kurylyk and Hayashi, 2016), but its accuracy is still reasonable (e.g., Klene et al., 2001; Shiklomanov and Nelson, 2002; Hrbáček and Uxa, 2020). Besides, its simplicity and low requirements for input data as compared to more complex analytical

or numerical models (e.g., Kudryavtsev et al., 1977; Jafarov et al., 2012; Westermann et al., 2016) is highly advantageous for palaeo-applications as fewer assumptions have to be made. Here, it is solved for a uniform, non-layered ground while ignoring any of its thaw-related mechanical responses.

  The PERICLIMv1.0 deduces the thawing-season temperature conditions in the above way, which it further converts into annual as well as freezing-season air temperature attributes based on the assumed annual air temperature range as detailed

below. Note that its code is implemented and disseminated as R package.

### 2.1   Model input variables

The model requires inputs on the palaeo-active-layer thickness $\xi$ [m], the volumetric ground moisture content $\phi$ [−], the dry ground bulk density $\rho$ [kg m$^{-3}$], the ground quartz content $q$ [−], the ground grain-size class $f/c$ [−], the ground-surface thawing $n$-factor $n_\mathrm{t}$ [−], and the annual air temperature range $A_\mathrm{a}$ [°C] (Table 1), which corresponds to the difference between

the mean air temperature of the warmest and coldest month. The ground physical properties are assumed to characterize the entire modelling domain (∼active layer) and are treated as constant over time.

### 2.2   Ground-surface and air thawing index

The Stefan equation for calculating the active-layer thickness in a homogeneous substratum with constant physical properties has the following form (Lunardini, 1981):

$$\xi = \sqrt{\frac{2k_\mathrm{t}I_\mathrm{ts}}{L\phi\rho_\mathrm{w}}}, \tag{1}$$

where $k_\mathrm{t}$ [W m$^{-1}$ K$^{-1}$] is the thermal conductivity of the thawed ground calculated here as a function of its dry bulk density and volumetric moisture content using the Johansen's (1977) thermal-conductivity model (Appendix A), $I_\mathrm{ts}$ [°C d] is the ground-surface thawing index defined as a sum of positive daily ground-surface temperatures in the thawing season (Fig. 1), $L$ [334 000 J kg$^{-1}$] is the specific latent heat of fusion of water, and $\rho_\mathrm{w}$ [1 000 kg m$^{-3}$] is the density of water. Note that $I_\mathrm{ts}$

**Table 1.** List of model input and output variables, their symbols, value ranges, and units.

| Variable | Symbol | Value range | Unit |
|---|---|---|---|
| **Inputs** | | | |
| Palaeo-active-layer thickness | $\xi$ | $\xi > 0$ | m |
| Volumetric ground moisture content | $\phi$ | $0 < \phi \le 1$ | – |
| Dry ground bulk density | $\rho$ | $0 < \rho \le 2700$ | $\mathrm{kg\,m^{-3}}$ |
| Ground quartz content | $q$ | $0 \le q \le 1$ | – |
| Ground grain-size class | $f/c$ | 'fine' or 'coarse' | – |
| Ground-surface thawing $n$-factor | $n_\mathrm{t}$ | $n_\mathrm{t} > 0$ | – |
| Annual air temperature range | $A_\mathrm{a}$ | $A_\mathrm{a} > 0$ | °C |
| **Outputs** | | | |
| Mean annual air temperature | MAAT | $-\frac{A_\mathrm{a}}{2} < \mathrm{MAAT} \le 0$ | °C |
| Mean air temperature of the warmest month | MATWM | $\mathrm{MATWM} > 0$ | °C |
| Mean air temperature of the coldest month | MATCM | $\mathrm{MATCM} < 0$ | °C |
| Mean air temperature of the thawing season | MATTS | $\mathrm{MATTS} > 0$ | °C |
| Mean air temperature of the freezing season | MATFS | $\mathrm{MATFS} < 0$ | °C |
| Air thawing index | $I_\mathrm{ta}$ | $I_\mathrm{ta} > 0$ | °C d |
| Air freezing index | $I_\mathrm{fa}$ | $I_\mathrm{fa} < 0$ | °C d |
| Length of the thawing season | $L_\mathrm{t}$ | $0 < L_\mathrm{t} < 365$ | d |
| Length of the freezing season | $L_\mathrm{f}$ | $0 < L_\mathrm{f} < 365$ | d |
| Ground-surface thawing index | $I_\mathrm{ts}$ | $I_\mathrm{ts} > 0$ | °C d |

must be multiplied by the scaling factor of $86\,400\,\mathrm{s\,d^{-1}}$ in Eq. (1) to obtain the active-layer thickness in meters. Besides, the product of $\phi$ and $\rho_\mathrm{w}$ can be alternatively substituted by that of the gravimetric ground moisture content and dry ground bulk density as their results are identical.

The ground-surface thawing index required to reach a given active-layer thickness can be obtained if Eq. (1) is rearranged such as:

$$I_\mathrm{ts} = \frac{\xi^2 L \phi \rho_\mathrm{w}}{2 k_\mathrm{t}}. \tag{2}$$

The ground-surface thawing index can then be converted into the air thawing index $I_\mathrm{ta}$ [°C d] through the ground-surface thawing $n$-factor (Lunardini, 1978), which is a simple empirical transfer function that has been widely used to parametrize the thawing-season air–ground temperature relations across permafrost landscapes (e.g., Klene et al., 2001; Gisnås et al., 2017):

$$I_\mathrm{ta} = \frac{I_\mathrm{ts}}{n_\mathrm{t}}. \tag{3}$$

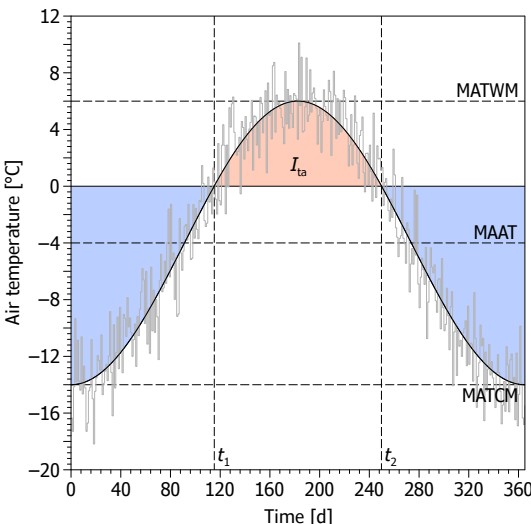

**Figure 1.** An idealized course of air temperature during the year described by a sine function with a mean of −4 °C and a range of 20 °C (black solid line) superimposed on the annual air temperature curve with daily variations (grey solid line). The air thawing index is shown in red, while the blue areas depict partial air freezing indices for the preceding (left) and subsequent (right) freezing season, respectively. See text and Table 1 for abbrevations.

Note that the effect of snow cover does not have to be accounted for because the Stefan equation (Eq. 2) considers solely the thawing-season temperatures responsible for the active-layer thawing. Likewise, the resulting air thawing index is later used only to calculate other air temperature characteristics and these are not affected by snow in any way. Such a scheme is particularly advantageous because it keeps the number of inputs small. Yet, it should be borne in mind that it is suitable especially for locations without long-lasting snow cover, which might disrupt the coupling between the air and ground temperatures during prolonged snow-melting periods (e.g., Gisnås et al., 2016).

## 2.3 Air temperature characteristics

The temporal evolution of air temperature over the year $T_{\mathrm{a}}(t)$ [°C] can be well described by a sine wave (Fig. 1) such as:

$$T_{\mathrm{a}}(t) = \mathrm{MAAT} + \frac{A_{\mathrm{a}}}{2}\sin\left(\frac{2\pi t}{P}\right), \tag{4}$$

where $t$ [d] is the time and $P$ [365 d] is the period of air temperature oscillations. Note that the annual air temperature range must be halved in Eq. (4) and the subsequent equations in order to characterize the annual temperature variations around MAAT (Fig. 1).

The air thawing index represents positive area under the annual air temperature curve (Fig. 1) and can be calculated by integrating Eq. (4) over the thawing season as follows:

$$I_{\mathrm{ta}} = \int_{t_1}^{t_2} T_{\mathrm{a}}(t)\,\mathrm{d}t, \tag{5}$$

with

$$t_1 = \arcsin\left(-\frac{\mathrm{MAAT}}{\frac{A_{\mathrm{a}}}{2}}\right)\frac{P}{2\pi}, \tag{6}$$

$$t_2 = \left[\pi - \arcsin\left(-\frac{\mathrm{MAAT}}{\frac{A_{\mathrm{a}}}{2}}\right)\right]\frac{P}{2\pi}, \tag{7}$$

where $t_1$ is the time when the air temperature curve crosses the zero-degree Celsius level from below ($\sim$thawing season be-
gins), while $t_2$ is the time when it crosses this level from above ($\sim$thawing season ends) (e.g., Nelson and Outcalt, 1987). Unfortunately, Eq. (5) has no analytical solution for MAAT. However, it can be derived from a nomogram (Fig. 2) or, as here, it can be calculated numerically using the bisection root-finding method searching for MAAT such that $-\frac{A_{\mathrm{a}}}{2} < \mathrm{MAAT} \leq 0$. This condition ensures that both positive and negative air temperatures have occurred during the year, which is an essential prerequisite for the active layer to form. Admittedly, it is simplistic because air–ground temperatures are modulated by sur-
face and subsurface offsets so that permafrost–seasonal frost boundary usually occurs at slightly negative MAAT (Smith and Riseborough, 2002). Consequently, there might be a risk that the model is incorrectly applied to seasonal-frost conditions. However, this can be easily prevented if periglacial features that have indisputably developed in the presence of permafrost are examined.

Once MAAT is known, the air freezing index $I_{\mathrm{fa}}$ [$^\circ$C d] can be simply computed as:

$$I_{\mathrm{fa}} = \mathrm{MAAT}\,P - I_{\mathrm{ta}}. \tag{8}$$

Furthermore, the mean air temperature of the warmest MATWM [$^\circ$C] and coldest MATCM [$^\circ$C] month is calculated as:

$$\mathrm{MATWM} = \mathrm{MAAT} + \frac{A_{\mathrm{a}}}{2}, \tag{9}$$

$$\mathrm{MATCM} = \mathrm{MAAT} - \frac{A_{\mathrm{a}}}{2}. \tag{10}$$

The mean air temperature of the thawing MATTS [$^\circ$C] and freezing MATFS [$^\circ$C] season is defined as:

$$\mathrm{MATTS} = \frac{I_{\mathrm{ta}}}{L_{\mathrm{t}}}, \tag{11}$$

$$\mathrm{MATFS} = \frac{I_{\mathrm{fa}}}{L_{\mathrm{f}}}, \tag{12}$$

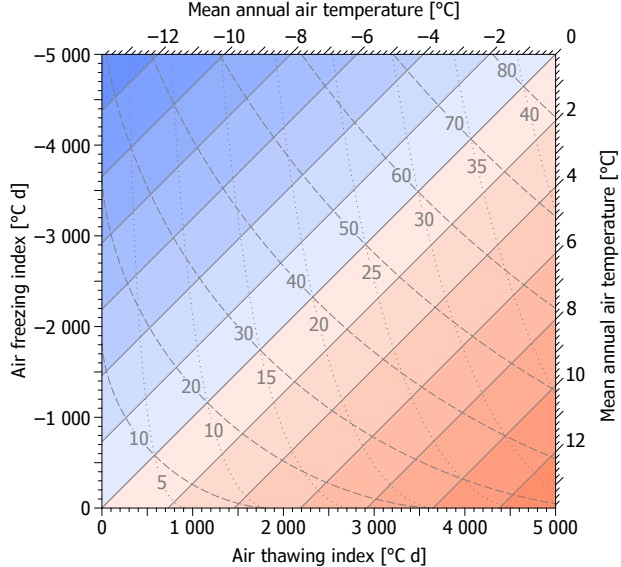

**Figure 2.** A nomogram showing relations between the air thawing and freezing index, mean annual air temperature (solid diagonal lines), annual air temperature range (dashed curved lines), and mean air temperature of the warmest month (dotted curved lines). Note that the value of the mean annual air temperature (read diagonally) and the air freezing index (read horizontally) is obtained at the intersection of the air thawing index (read vertically) and the annual air temperature range or the mean air temperature of the warmest month (both read curvilinearly).

where $L_t$ [d] and $L_f$ [d] is the duration of the thawing and freezing season, respectively, which is expressed from Eq. (6) and (7) as:

$$L_t = t_2 - t_1 = \left[ \pi - 2\arcsin\left( -\frac{\text{MAAT}}{\frac{A_a}{2}} \right) \right] \frac{P}{2\pi}, \tag{13}$$

$$L_f = P - L_t. \tag{14}$$

Unconventionally, the model can also be solved with the range of annual air temperature oscillations defined by MATWM (cf. Williams, 1975) if its difference from MAAT (i.e., MATWM − MAAT) is substituted for $\frac{A_a}{2}$ in Eq. (4) and elsewhere (Appendix B). Unsurprisingly, but importantly, solutions based on alternate model input variables, as suggested above, produce identical outputs, allowing the model adaptations to specific situations and available data.

**3   Model validation**

Performance of the model was tested using the same simulation schemes as for palaeo-applications detailed in the next section (see Sect. 4), but on the basis of data from modern permafrost environments of the James Ross Island and the Alaskan Arctic (Appendix C). Comparisons of the modelled and observed data showed relatively good agreements and clustering along the

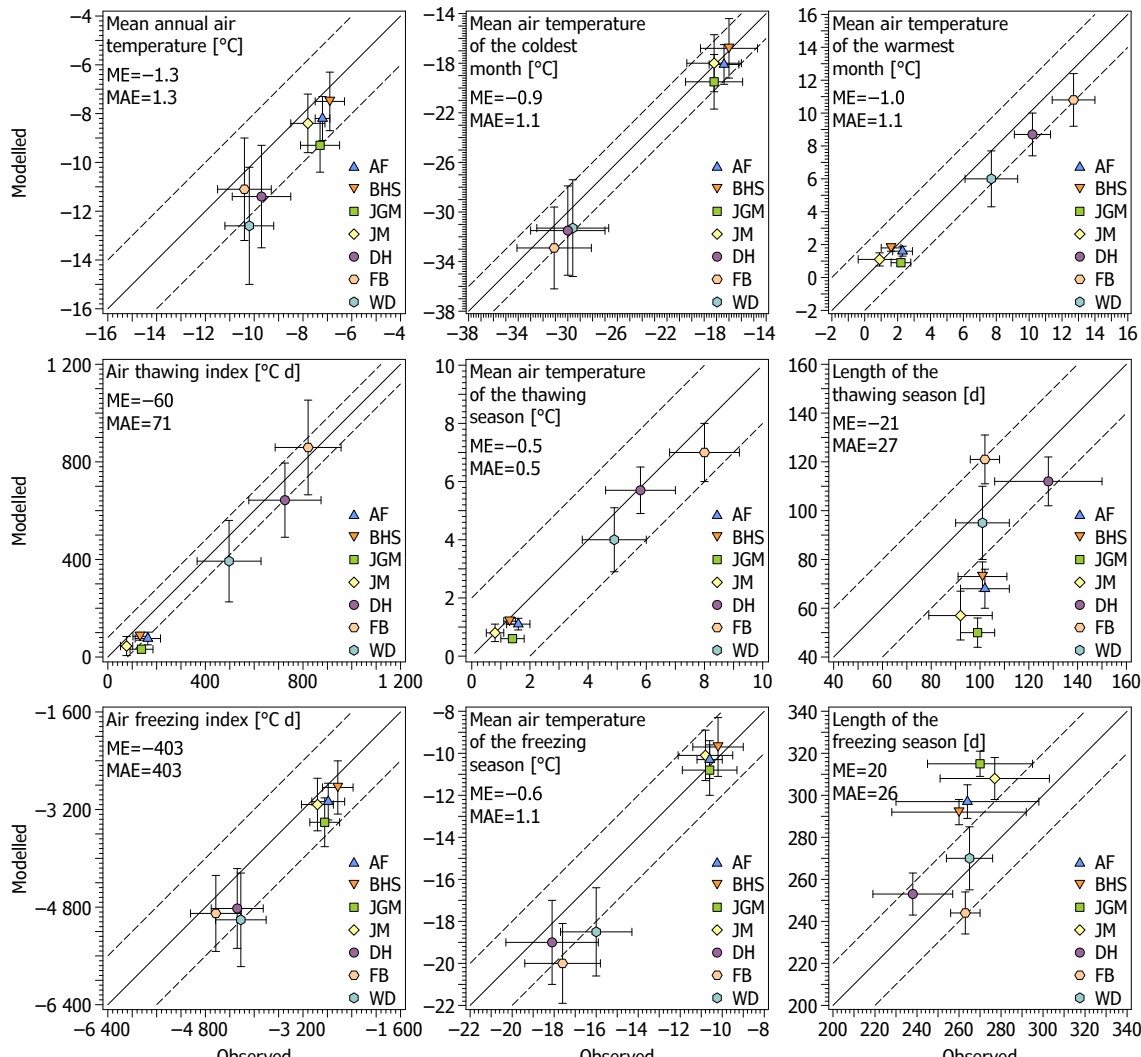

**Figure 3.** Means and standard deviations of observed and modelled air temperature characteristics at the James Ross Island and Alaskan Arctic validation sites. Acronyms ME and MAE under the plot labels correspond to the site-weighted mean error and the site-weighted mean absolute error, respectively, while those at the bottom right of the plots indicate the names of the validation sites: Abernethy Flats (AF), Berry Hill slopes (BHS), Johann Gregor Mendel (JGM), Johnson Mesa (JM), Deadhorse (DH), Franklin Bluffs (FB), and West Dock (WD).

lines of equality for most air temperature characteristics (Fig. 3), which suggests that the model might also work well over a
wider range of climates. Generally, however, the model underestimated the results, with those from James Ross Island being
somewhat more accurate and less scattered than those from Alaska (Fig. 3).

MAAT exhibited an average mean error and an average mean absolute error of −1.3 °C and 1.3 °C, respectively (Fig. 3).
Mean absolute error was ≤ 1 °C and ≤ 2 °C at 57 % and 86 % of the validation sites, respectively, and its maximum was 2.4 °C.

MATWM and MATCM showed slightly lower bias as their average mean errors attained $-1.0\,°\text{C}$ and $-0.9\,°\text{C}$, respectively, and average mean absolute errors achieved $1.1\,°\text{C}$ for both characteristics (Fig. 3). Their mean absolute deviations were $\leq 1\,°\text{C}$ at 43 % of the validation sites and $\leq 2\,°\text{C}$ at 100 % of them. Mean absolute MATWM and MATCM errors reached a maximum of $1.9\,°\text{C}$ and $1.8\,°\text{C}$, respectively.

$I_{\text{ta}}$ and $I_{\text{fa}}$ were modelled with average mean errors of $-60\,°\text{C}\,\text{d}$ and $-403\,°\text{C}\,\text{d}$, respectively, which corresponds to $-16\,\%$ and $-12\,\%$ of the observed average mean values, and their average mean absolute errors reached $71\,°\text{C}\,\text{d}$ and $403\,°\text{C}\,\text{d}$, respectively (Fig. 3). $I_{\text{ta}}$ biased by $\leq 40\,°\text{C}\,\text{d}$ at 29 % of the validation sites and by $\leq 80\,°\text{C}\,\text{d}$ at 43 % of them. $I_{\text{fa}}$, which achieves one order of magnitude larger values, deviated by $\leq 400\,°\text{C}\,\text{d}$ at 57 % of the validations and by $\leq 800\,°\text{C}\,\text{d}$ at 100 % of them. Maximum mean absolute departures of $I_{\text{ta}}$ and $I_{\text{fa}}$ achieved $106\,°\text{C}\,\text{d}$ and $789\,°\text{C}\,\text{d}$, respectively.

MATTS exhibited an average mean error and an average mean absolute error of $-0.5\,°\text{C}$ and $0.5\,°\text{C}$, respectively, while for MATFS the mean errors averaged $-0.6\,°\text{C}$ and $1.1\,°\text{C}$ (Fig. 3). The agreement between the modelled and observed MATTS was $\leq 1\,°\text{C}$ at 100 % of the validation sites, with maximum mean absolute error of $1.0\,°\text{C}$. MATFS deviated by $\leq 1\,°\text{C}$ at 71 % of the validation sites, and at worst, it was $2.5\,°\text{C}$.

Since $L_{\text{t}}$ and $L_{\text{f}}$ inherently counteract, their characteristics mirror each other if the values are not rounded. $L_{\text{t}}$ tended to be underestimated by an average of $21\,\text{d}$, while $L_{\text{f}}$ was overestimated by an average of $20\,\text{d}$ (Fig. 3), which comprised $-20\,\%$ and $8\,\%$ of the observed average mean $L_{\text{t}}$ and $L_{\text{f}}$, respectively, and their average mean absolute errors achieved $27\,\text{d}$ and $26\,\text{d}$. Mean underestimation or overestimation was $\leq 20\,\text{d}$ at 43 % of the validation sites and $\leq 40\,\text{d}$ at 86 %. Maximum mean deviation was up to $49\,\text{d}$.

## 4   Model application

As a feasibility study, the model was utilized experimentally for derivation of palaeo-air temperature conditions on the basis of relict cryoturbation structures. Cryoturbations ($\sim$periglacial involutions) are characterized by folded and/or dislocated strata of unconsolidated sediments caused by recurrent freeze-thaw-induced processes operating within the active layer over permafrost, which limits the vertical extent of the cryoturbations from below and also acts as an impervious boundary that provokes well saturated conditions. As such, cryoturbations are thought to indicate the thickness of the active layer as well as the presence of permafrost at the time of their development. Also, MAAT thresholds of $<-8\,°\text{C}$ and $<-4\,°\text{C}$ have been suggested for their formation within coarse- and fine-grained substrates, respectively (Vandenberghe, 2013; French, 2017), the validity of which can be assessed with the model as well.

### 4.1   Study sites

We consider two study sites in the Czech Republic where vertical cross-sections through cryoturbated horizons, portraying the palaeo-active layers (Fig. 4), were exposed and sampled for those attributes that allowed to define the most plausible ranges of the model input variables.

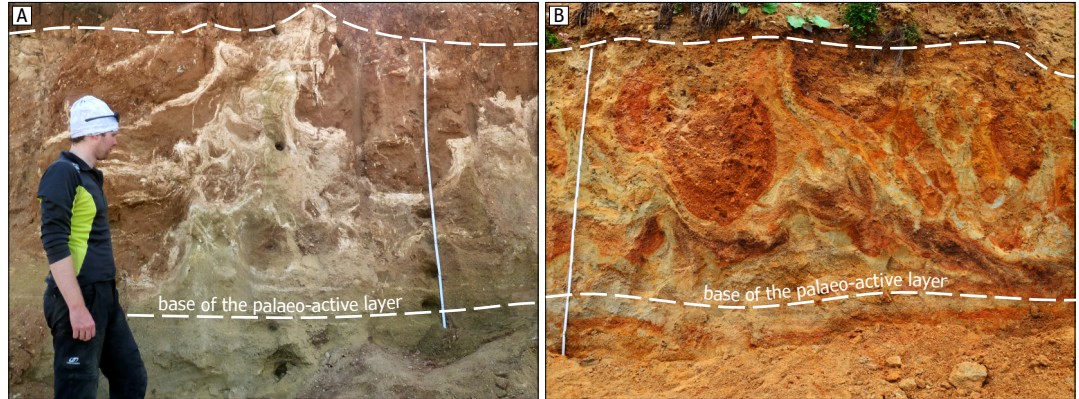

**Figure 4.** Cross-sections through cryoturbation structures at the (A) Brno–Černovice and (B) Nebanice site with the white dashed lines indicating their vertical extent and thereby the thickness of the palaeo-active layer. Note that the sections above the suggested base of the palaeo-active layer show distinct involution structures caused by freeze-thaw-induced processes, while those below have largely retained their primary structures with horizontal to sub-horizontal bedding planes.

The Brno–Černovice site (49°10′43″ N, 16°38′56″ E, 240 m asl) is an active sand–gravel pit situated about 3 km southeast of the centre of Brno, South Moravia. The mine is embedded within sands and gravels of the Tuřany terrace of the Svitava River, in reality a highly flattened alluvial fan, tentatively attributed to the Günz glaciation (∼Marine Isotope Stage [MIS] 22–11), which is located about 1.5 km east of the present channelized river bed and ∼40 m higher. The material tends to coarsen down to a depth of 6–13.5 m, under which the Neogene clayey sands and gravels emerge (Musil et al., 1996; Musil, 1997; Bubík et al., 2000). The cryoturbations mostly consist of sands to gravels that constitute ∼80 % and ∼14 % of the substrate, respectively, and are deformed by numerous injection tongues (Fig. 4). Currently, the cryoturbations rest ∼1.6–3.6 m under the ground surface, but we suppose they were just below it when they developed.

The Nebanice site (50°07′13″ N, 12°27′09″ E, 430 m asl) is a currently inactive sand–gravel pit about 1.5 km west of the centre of Nebanice, West Bohemia. It is set in sands and gravels of the Nebanice terrace of the Ohře River, which is thought to have originated during the Mindel–Riss glaciations (∼MIS 10), and now is situated about 200 m north of the river bed at a relative height of ∼10 m (Šantrůček et al., 1994; Balatka et al., 2019). The terrace sediments have a thickness of 1.7–3.8 m (Balatka et al., 2019) and are underlain by the Pliocene–Lower Pleistocene clays, sands, and gravels (Špičáková et al., 2000). The cryoturbations are mostly composed of sands and gravels that comprise ∼67 % and ∼29 % of the material, respectively, and take the form of festoons and ball-and-pillow structures (Fig. 4) emerging in the uppermost part of the terrace ∼1.1–2.8 m below the modern ground surface as they were later overlaid by younger sediments and soils as well.

Cryoturbations, like other permafrost-related features, in the area of the Czech Republic have been tentatively attributed to the last glacial period (Czudek, 2005). We believe that the studied relict cryoturbations indicate past environmental conditions similar to those at the end of the Last Glacial Maximum (LGM) when analogous features formed in nearby regions (Kasse, 1993; Huijzer and Vandenberghe, 1998; Kasse et al., 2003; Bertran et al., 2014).

**Table 2.** Variables used to model palaeo-air temperature characteristics related to the cryoturbations at the Brno–Černovice and Nebanice site. Note that the superscripts *norm*, *beta*, and *unif* indicate normal, beta, and uniform distributions describing the respective variables.

| Site | $\xi$ [m]$^{norm}$ | $\phi$ [−]$^{beta}$ | $\rho$ [kg m$^{-3}$]$^{norm}$ | $q$ [−]$^{unif}$ | $f/c$ | $n_t$ [−]$^{norm}$ | $A_a$ [°C]$^{norm}$ |
|---|---|---|---|---|---|---|---|
| Brno–Černovice | 1.58±0.28 | 0.088–0.394 | 1635±120 | 0.30–0.56 | coarse | 1.03±0.12 | 23.2±2.4 to 33.2±2.4 |
| Nebanice | 1.40±0.13 | 0.114–0.391 | 1645±116 | 0.30–0.57 | coarse | 1.03±0.12 | 20.9±2.6 to 30.9±2.6 |

See Table 1 for abbrevations.

## 4.2 Model set-up

### 4.2.1 Palaeo-active-layer thickness

The palaeo-active-layer thickness was considered to be represented by the vertical extent of the cryoturbated horizons (Fig. 4), which was determined in horizontal steps of 0.2 m along the entire length of each cross-section (4.4 and 8.6 m), giving 22 and 43 measurements that averaged 1.58±0.28 and 1.40±0.13 m at the Brno–Černovice and Nebanice site, respectively (Table 2).

### 4.2.2 Ground physical properties

Ground physical properties were set using intact samples collected at four representative positions along four vertical profiles within each cross-section (16 samples per cross-section) into 100 cm$^3$ stainless steel cylinders, which were then weighted in wet and dry states as well as sieved to assess their volumetric moisture content, dry bulk density, and texture. The current volumetric moisture content averaged 8.8 % and 11.4 % at the Brno–Černovice and Nebanice site, respectively, and these values were assumed to be the lower moisture thresholds. The upper moisture thresholds were supposed to be given by average ground porosity, which was calculated as a function of the dry bulk density (Eq. A4), and achieved 39.4 % and 39.1 %, respectively (Table 2). Because cryoturbations require well saturated conditions for at least a part of the thawing season for their development (Vandenberghe, 2013; French, 2017), we further skewed the moisture-content values leftwards using a beta distribution as follows:

$$\phi(x) = \phi_0 + f(x; \alpha, \beta)(n - \phi_0), \tag{15}$$

where $\phi_0$ [−] is the current average volumetric ground moisture content, $f(x; \alpha, \beta)$ expresses the probability density function of the beta distribution for $0 \le x \le 1$ with shape parameters tentatively assumed to be $\alpha = 5$ and $\beta = 2$ that yield a mode of 0.8, and $n$ [−] is the average ground porosity. The resulting values are thus bounded by the lower and upper moisture thresholds (Table 2), but show left-skewed distributions peaking at ~33.3 % and ~33.6 % at the Brno–Černovice and Nebanice site, respectively, which corresponds to the degree of saturation of ~84.5 % and ~85.8 % (Fig. 5). Since the vast majority of moisture in sands and gravels tends to undergo phase changes (Andersland and Ladanyi, 2004) and its unfreezing portion is supposed to influence the calculations negligibly if its ratio to the total moisture content is at levels up to several tens of percent (Uxa, 2017), the moisture contents were not further adjusted for an unfrozen moisture. Slightly lower moisture content

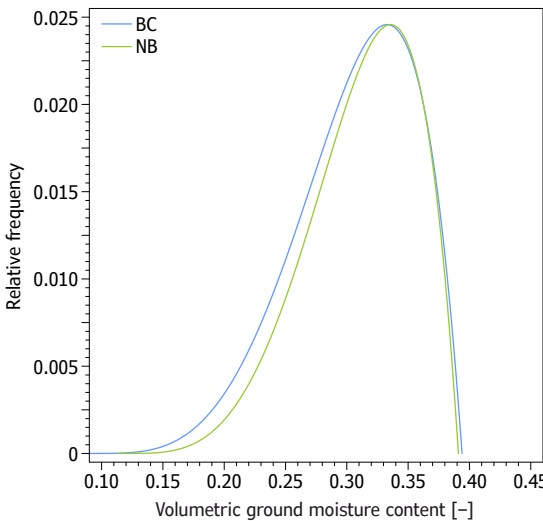

**Figure 5.** Probability distributions of volumetric ground moisture contents assumed for the Brno–Černovice (BC) and Nebanice (NB) site.

of the eastern site in Brno–Černovice as compared to Nebanice in the west (Fig. 5) is likely to be reasonable because it can be understood as including the continentality and elevation effects as well as the thicker palaeo-active layer (Table 2) in which the same amount of water has a lower volume fraction.

Other ground physical properties were supposed to be unchanged since the cryoturbations developed. The dry ground bulk density was $1635{\pm}120\,\mathrm{kg\,m^{-3}}$ and $1645{\pm}116\,\mathrm{kg\,m^{-3}}$ at the Brno–Černovice and Nebanice site, respectively. The ground quartz content was estimated at 30–56 % and 30–57 %, respectively, on the basis of the proportions of clay–silt and sand fractions (see Appendix A). Given the low proportion of clay–silt fraction and the presence of gravel (see Sect. 4.1), the substrates were treated as coarse-grained (sensu Johansen, 1977) at both sites (Table 2).

### 4.2.3  Ground-surface thawing *n*-factor

Since the coupling between the air and ground-surface temperatures is principally governed by ground-surface cover and its properties themselves (Westermann et al., 2015), the ground-surface thawing *n*-factors were estimated on the basis of values published elsewhere for bare (excluding bedrock and debris) to little vegetated (sparse lichens, mosses, or grasses) surfaces, which are assumed to have existed at the study sites when the cryoturbations originated because treeless landscapes dominated in South Moravia and West Bohemia at that time (e.g., Kuneš et al., 2008) and also because the cryoturbated horizons contain no or negligible amounts of organic remains. We took over a total of forty-one ground-surface thawing *n*-factor values reported from mid-latitude and mostly permafrost regions of Central-Eastern Norway (Juliussen and Humlum, 2007) and British Columbia/Yukon and Labrador, Canada (Lewkowicz et al., 2012; Way and Lewkowicz, 2018) that are believed to be representative for the study sites as those locations also occur outside the Arctic Polar Circle and, as such, have comparable insolation budgets owing to the absence of polar-day periods, which might impose an undesirable growth of the ground-surface

thawing *n*-factor values (e.g., Shur and Slavin-Borovskiy, 1993). Overall, the collected *n*-factors averaged 1.03±0.12, which was assigned to both study sites (Table 2).

### 4.2.4 Annual air temperature range

As a starting point, we used the annual air temperature ranges based on monthly air temperatures measured in 1981–2010 at meteorological stations located about 4 and 7 km southeast and southwest of the Brno–Černovice and Nebanice site, respectively, at elevations of 241 and 475 m asl (Czech Hydrometeorological Institute, 2020), which averaged 23.2±2.4 and 20.9±2.6 °C, respectively (Table 2). Additionally, we also assumed stepwise perturbations of 2 °C to these annual air temperature ranges up to 33.2±2.4 and 30.9±2.6 °C, respectively, which is within the range of 28–36 °C suggested by previous regional estimates for the end of the LGM (Huijzer and Vandenberghe, 1998) and at the same time maintains the contrasts between the study sites (Table 2).

### 4.2.5 Simulations

The model was solved by the Monte Carlo simulation with Latin hypercube sampling (LHS) (McKay et al., 1979) using *lhs* R package (Cannel, 2020), which discretizes the probability density functions of the individual model input variables (Table 2) into $N$ non-overlapping intervals of equal probability $1/N$ and then randomly selects one value from each. Subsequently, these samples are randomly matched and used as uncorrelated inputs for $N$ model runs (McKay et al., 1979). LHS is computationally more efficient than simple random sampling because its sampling scheme adapts to the probability density functions of the input variables, and thus it requires fewer simulations to achieve stable outputs if $N$ is large enough.

We realized 1 000 model runs for six scenarios of the annual air temperature ranges at each study site, of which 78.1–91.1 % produced physically feasible combinations of the input variables and provided the most plausible palaeo-air temperature characteristics that account for the uncertainty as well as natural temporal variability of the inputs.

### 4.3 Modelled palaeo-air temperature characteristics

Most modelled palaeo-air temperature characteristics changed in various extents depending on the scenarios of the annual air temperature ranges, the higher values of which generally caused colder and/or more continental conditions as well as larger scatters of the model outputs. This was especially true for MAAT and the freezing-season characteristics, whereas the changes were comparatively milder for the thawing-season attributes (Fig. 6).

MAAT was modelled at −6.6±2.7 to −3.3±2.0 °C and −7.0±1.9 to −3.2±1.5 °C at the Brno–Černovice and Nebanice site, respectively (Fig. 6), which corresponds to its average decline of −16.0 to −12.7 °C and −15.1 to −11.3 °C, respectively, in comparison with the 1981–2010 period. The average contrasts between the warmest and coldest MAAT scenarios based on the current and assumed past annual air temperature ranges, respectively, were 3.3 °C and 3.8 °C at the Brno–Černovice and Nebanice site, respectively, which means that MAAT changed by 33 % and 38 % of the change in the annual air temperature range.

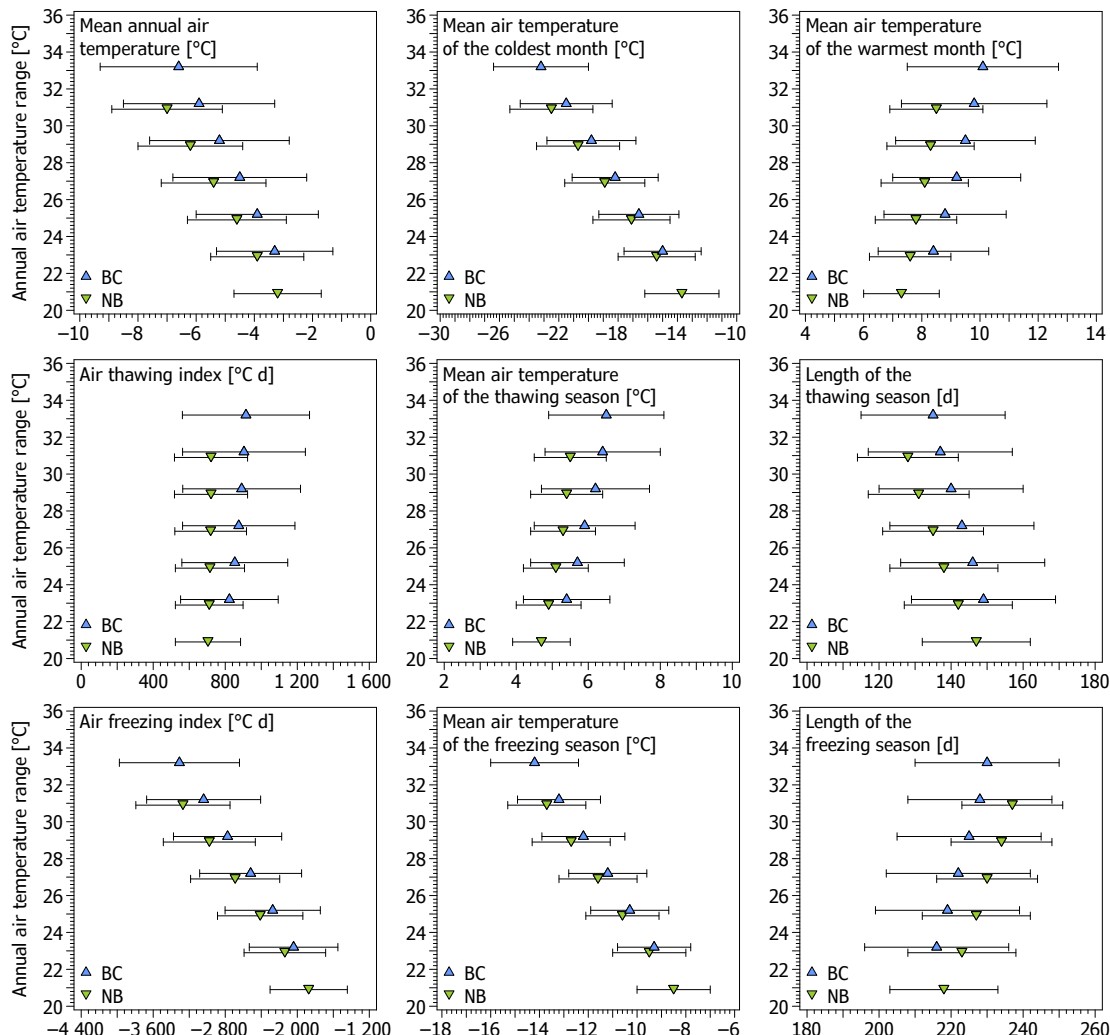

**Figure 6.** Means and standard deviations of modelled palaeo-air temperature characteristics at the Brno–Černovice (BC) and Nebanice (NB) site for six scenarios of the annual air temperature ranges.

The thawing seasons had MATTS of 5.4±1.2 to 6.5±1.6 °C and 4.7±0.8 to 5.5±1.0 °C at the Brno–Černovice and Nebanice site, respectively, and culminated with MATWM of 8.4±1.9 to 10.1±2.6 °C and 7.3±1.3 to 8.5±1.6 °C (Fig. 6), which suggests that MATTS changed on average by 11 % and 8 % of the magnitudes of the annual air temperature range perturbations, while MATWM varied on average by 17 % and 12 %. $I_{\text{ta}}$ showed even more consistent outputs as it attained 823±271 to 915±353 °C d and 704±181 to 721±203 °C d at the Brno–Černovice and Nebanice site, respectively (Fig. 6), and, as such, it varied by as low as ∼1.1 % and ∼0.2 % per 1 °C change in the annual air temperature range. Likewise, $L_{\text{t}}$ was modelled at 135±20 to 149±20 d and 128±14 to 147±15 d (Fig. 6), which yields the respective changes of ∼1.0 % and ∼1.5 %.

The freezing seasons exhibited the largest scatter as well as the highest variability among the scenarios with MATFS of $-14.2\pm1.8$ to $-9.3\pm1.5\,°C$ and $-13.7\pm1.6$ to $-8.5\pm1.5\,°C$ at the Brno–Černovice and Nebanice site, respectively, and MATCM as low as $-23.2\pm3.2$ to $-15.0\pm2.6\,°C$ and $-22.5\pm2.8$ to $-13.7\pm2.5\,°C$ (Fig. 6). It thus follows that their variations reached on average 49 % and 52 % of the annual air temperature range perturbations for MATFS and as much as 82 % and 88 % for MATCM. $I_{fa}$ spanned $-3309\pm667$ to $-2041\pm492\,°C\,d$ and $-3270\pm523$ to $-1873\pm429\,°C\,d$ at the Brno–Černovice and Nebanice site, respectively (Fig. 6), which corresponds to average variations of $\sim6.2$ % and $\sim7.5$ % per $1\,°C$ change in the annual air temperature range. On the other hand, $L_f$ was modelled at $216\pm20$ to $230\pm20\,d$ and $218\pm15$ to $237\pm14\,d$, respectively, and thus it varied on average by as low as $\sim0.6$ % and $\sim0.9$ % of the magnitudes of the annual air temperature range perturbations.

## 5 Discussion

### 5.1 Model performance and limitations

Generally, the model validation using the modern data (Sect. 3) showed its relatively high accuracy for most air temperature characteristics if suitable input variables are available. It is remarkable, though, given that active-layer thickness commonly exhibits rather moderate to low correlations with annual or freezing-season air and ground temperature attributes, but on the other hand, it mostly strongly couples with thawing-season air and ground temperature indices (e.g., Frauenfeld et al., 2004; Åkerman and Johansson, 2008; Wu and Zhang, 2010). Since the model builds on active-layer thickness–thawing-season temperature relations, which it further converts into annual and freezing-season air temperature characteristics, this scheme gives rise to its reasonable accuracy. It also needs to be stressed that the model was successful regardless the ground physical properties used have certainly undergone at least slight changes since sampling and varied over the validation period. More accurate and less scattered outputs at the James Ross Island sites were largely due to a rather homogeneous distribution of ground physical properties within the active layer there (Hrbáček et al., 2017a; Hrbáček and Uxa, 2020). By contrast, the Alaskan profiles have a two-layer composition with a peat over mineral soil (Zhang, 1993). Besides, the model parameterizes the air temperature behaviour with a sine wave, which simplifies its actual evolution over the year and completely ignores sub-annual variations.

Overall, however, the model underestimated most air temperature characteristics (Fig. 3). The underestimation is attributed to intrinsic shortcomings of the Stefan equation (Eq. 2) that tends to deviate inversely proportional to the moisture content in the active layer as well as the active-layer temperature at the onset of thawing (Romanovsky and Osterkamp, 1997; Kurylyk and Hayashi, 2016). Also, it is associated with the Johansen's (1977) thermal-conductivity model (Appendix A), which tended to produce too high thermal conductivity values at the James Ross Island sites. Surely, the Stefan equation (Eq. 2) might be improved by a number of correction factors, but these require additional inputs, such as frozen thermal conductivity, thawed and frozen volumetric heat capacity, or active-layer temperature at the start of its thawing (Kurylyk and Hayashi, 2016). As such, the corrections are frequently difficult to implement even in many present-day situations and definitely are much less viable for palaeo-applications. Moreover, their inverse solution would not be straightforward and would probably demand iterative

techniques. Similarly, the Johansen's (1977) thermal-conductivity model is advantageous in that it requires fewer inputs as
compared with other solutions while having comparable accuracy (e.g., Dong et al., 2015; He et al., 2017; Zhang et al., 2018).
As such, it is also difficult to replace by another thermal-conductivity schemes.

Besides, it should be highlighted that for the model validation cases the active-layer thickness was reduced by the depth of
the temperature sensor, which was used to determine the ground-surface thawing $n$-factor (see Appendix C). However, if this
treatment was not done, the model outputs improved, with MAAT, MATWM, MATCM, MATTS, and MATFS showing average
errors $\leq 1\,°C$, $I_{ta}$ and $I_{fa}$ deviating on average by $-1\,\%$ and $-8\,\%$, respectively, and $L_t$ and $L_f$ by $-16\,\%$ and $6\,\%$, respectively,
because this compensated the intrinsic deviations of the Stefan equation (Eq. 2) described in the previous paragraph. Since
the published ground-surface thawing $n$-factors used for the palaeo-applications (Juliussen and Humlum, 2007; Lewkowicz et
al., 2012; Way and Lewkowicz, 2018) built on various near-surface depths of ground temperature sensors, we did not adjust
the active-layer thickness in these cases, and the modelled palaeo-air temperature characteristics could thus have a slightly
increased accuracy as well.

## 5.2 Comparison to previous palaeo-air temperature reconstructions

The palaeo-MAAT modelled for two sites in the Czech Republic (Sect. 4.3) was between $-7.0\pm1.9\,°C$ and $-3.2\pm1.5\,°C$ and
its corresponding reduction between $-16.0\,°C$ and $-11.3\,°C$ in comparison with the 1981–2010 period, which is relatively well
consistent with earlier reconstructions utilizing various relict periglacial features in Central European lowlands that suggested
MAAT depressions mostly between $-17\,°C$ and $-12\,°C$ (Poser, 1948; Büdel, 1953; Kaiser, 1960; Frenzel, 1967; Goździk,
1973; Huijzer and Vandenberghe, 1998; Marks et al., 2016). By contrast, it disagrees with slightly milder MAAT reductions of
at least $-7\,°C$ and $-13$ to $-6\,°C$ derived from groundwater data (Corcho Alvarado et al., 2011) and borehole temperature logs
(Šafanda and Rajver, 2001), but these may not necessarily correspond to the lowest temperatures because groundwater cycling
has been slowed or interrupted by permafrost, while ground temperature history may have partly been masked by latent-heat
effects. Similarly, the modelled MAAT differs rather highly from simulations of three Global Climate Models (GCMs), namely
Community Climate System Model 4, Model for Interdisciplinary Research on Climate–Earth System Model, and Max Planck
Institute Earth System Model Paleo, capturing the LGM at $\sim 22\,ka$, which were originally released by the Coupled Model
Intercomparison Project 5 in coarse resolutions, but later downscaled to $2'30''$ and made available via the WorldClim 1.4
dataset at https://www.worldclim.org (Hijmans et al., 2005). The GCMs suggest that MAAT was between $-4.5\,°C$ and $-0.4\,°C$
at the study sites, which corresponds to its reduction of between $-12.6\,°C$ and $-9\,°C$ in comparison with the 1981–2010 period.
Such relatively high temperatures, however, contrast with many proxy records and are thus suspected to be unrepresentative of
the coldest LGM conditions when continuous permafrost presumably occurred there (Vandenberghe et al., 2014; Lindgren et
al., 2016).

The modelled MATWM of $7.3\pm1.3$ to $10.1\pm2.6\,°C$ is well in the range of proxy-based MATWM of 5 to $13\,°C$ that has
been reconstructed for Central European lowlands (Huijzer and Vandenberghe, 1998; Marks et al., 2016). By contrast, it devi-
ates greatly from the GCMs outputs being as high as 11.1 to $18\,°C$. The modelled MATCM ranged widely from $-23.2\pm3.2$ to
$-13.7\pm2.5\,°C$, which, however, also pertains to other proxy records that show somewhat lower values of $-27$ to $-16.5\,°C$ (Hui-

jzer and Vandenberghe, 1998; Marks et al., 2016). The GCMs-based MATCM exhibits less variability of −21.6 to −17.7 °C, but its average is generally well consistent with that modelled by this study. Unfortunately, other modelled palaeo-air temperature characteristics cannot be validated directly because no reliable proxy records or GCMs outputs are available for them. However, as they are closely related to MAAT, MATWM, or MATCM, we believe that those attributes have similar plausibility.

Lastly, the modelled MAAT between −7.0±1.9 °C and −3.2±1.5 °C is relatively consistent with the MAAT threshold of <−4 °C commonly suggested for the formation of cryoturbation structures within fine-grained substrates (Vandenberghe, 2013). At the study sites, however, the cryoturbations consist of coarse-grained materials, for which the MAAT threshold as low as <−8 °C has been proposed (Vandenberghe, 2013). This study thus raises questions about the validity of the previously suggested MAAT thresholds for cryoturbation structures (see Vandenberghe, 2013; French, 2017) and calls for their thorough revision.

### 5.3 Sensitivity analysis

Global sensitivity analysis using multiple regression suggested that the palaeo-active-layer thickness and annual air temperature range had a major impact on the modelled palaeo-air temperature characteristics at the Brno–Černovice and Nebanice site (Fig. 7). The palaeo-active-layer thickness importantly showed the highest values of the standardized regression coefficients (SRCs) especially for the annual and thawing-season air temperature attributes. Similarly, the annual air temperature range also highly influenced MAAT and, in particular, it had the utmost control over the freezing-season air temperatures as its SRCs even tended to outweigh those for the palaeo-active-layer thickness at that time of the year. It thus follows that the freezing-season characteristics may have limited accuracy if the annual air temperature range is uncertain, which indeed translates into their higher variance (Fig. 6). On the other hand, the annual air temperature range slightly affected the thawing-season air temperature attributes (Fig. 7), and these are thus assumed to be the most plausible. Ground-surface and subsurface input variables, such as volumetric ground moisture content, ground dry bulk density, and ground-surface thawing $n$-factor, had considerably lower, albeit stable, influences on most modelled palaeo-air temperature characteristics. Ground quartz content was the weakest of the input variables as it was responsible only for a minor variability in the model outputs (Fig. 7).

### 5.4 Model applicability to periglacial features

Besides cryoturbations, we assume that the model could be utilized for deriving palaeo-air temperature characteristics on the basis of any relict periglacial features that can indicate former active-layer thickness, such as some kinds of patterned ground, some solifluction structures, frost-wedge tops, autochthonous blockfields, mountain-top detritus, active-layer detachment slides, up-frozen clasts, indurated horizons, or frost-weathering microstructures (Ballantyne and Harris, 1994; Matsuoka, 2011; Ballantyne, 2018). Also, it could be easily adapted for seasonal-frost features, although the estimation of snow conditions would be complicated.

Since most periglacial features develop on at least decadal or centennial timescales (e.g., Karte, 1983; Matsuoka, 2001; Ballantyne, 2018), inherently involving natural variations in climate as well as active-layer thickness, we hypothesize that their vertical extent frequently also includes the bottom transient layer where the boundary between the active layer and permafrost

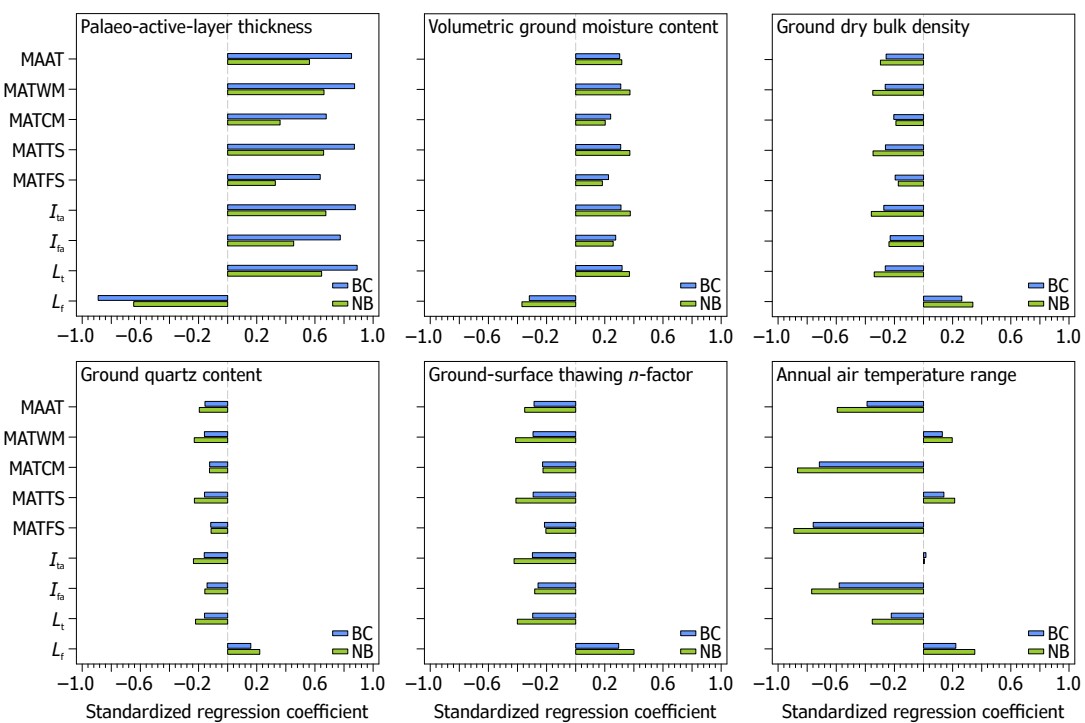

**Figure 7.** Sensitivity of the modelled palaeo-air temperature characteristics at the Brno–Černovice (BC) and Nebanice (NB) site to individual model input variables expressed by standardized regression coefficients. See text and Table 1 for abbrevations.

has fluctuated when periglacial features formed (cf. Shur et al., 2005). This implies that the palaeo-active-layer thickness may appear as a dispersed rather than a sharp boundary in many vertical cross-sections. Special attention must thus be paid to avoid any ambiguity in both the identification of relict periglacial features and the determination of the palaeo-active layer that may be severely degraded. Indeed, some periglacial features may be produced by seasonal frost alone, while some look-

390 alike features may even have a non-periglacial origin (Ballantyne and Harris, 1994; Ballantyne, 2018). Nonetheless, even if identified correctly, problems may arise in places where ground-surface level has changed over time due to sedimentation or erosion. Besides, high uncertainty can probably be expected especially for periglacial features composed of pebbly to bouldery materials, such as blockfields or mountain-top detritus, in which the vertical variability of ground physical properties is extremely high (Ballantyne, 1998) that is difficult to describe by single-value variables. On slopes, these materials may also

provoke non-conductive heat-transfer processes, which give rise to high-magnitude variations in ground temperatures over short distances that cannot be addressed by simple heat conduction models (Wicky and Hauck, 2017).

Given the above considerations, the modelling should ideally utilize coexisting periglacial features for more robust palaeo-air temperature estimates. Admittedly, it can capture only a snapshot of the temperature history, but this is no different from numerous other palaeo-indicators, such as various glacial deposits. Moreover, if relict periglacial assemblages of different ages

occur in a given region, they may eventually provide a more complete record of former temperature conditions. Undoubtedly,

dating of periglacial features is still challenging, because they can have a highly complex formation history, which partly devaluates their palaeo-environmental importance. Nonetheless, this shortcoming is increasingly being suppressed by improved dating methods that bring more reliable periglacial chronologies (e.g., Andrieux et al., 2018; Nyland et al., 2020; Engel et al., 2020).

## 6   Conclusions

The PERICLIMv1.0 is a novel easy-to-use model that derives palaeo-air temperature characteristics related to the palaeo-active-layer thickness, which can be recognized using many relict periglacial features found in past permafrost regions. The model evaluation against modern temperature records demonstrated that it reproduces air temperature characteristics, such as MAAT, MATWM, MATCM, MATTS, or MATFS, with average errors $\leq 1.3\,^{\circ}\mathrm{C}$. Besides, $I_{\mathrm{ta}}$ and $I_{\mathrm{fa}}$ deviates on average by $-16\,\%$ and $-12\,\%$, respectively, while $L_{\mathrm{t}}$ and $L_{\mathrm{f}}$ tends to be on average underestimated and overestimated by $-20\,\%$ and $8\,\%$, respectively. The palaeo-MAAT modelled for two sites in the Czech Republic hosting relict cryoturbation structures was between $-7.0\pm1.9\,^{\circ}\mathrm{C}$ and $-3.2\pm1.5\,^{\circ}\mathrm{C}$ and its corresponding reduction was between $-16.0\,^{\circ}\mathrm{C}$ and $-11.3\,^{\circ}\mathrm{C}$ in comparison with the 1981–2010 period, which is relatively well in line with earlier reconstructions utilizing various palaeo-archives.

These initial results are promising and suggest that the model could become a useful tool for reconstructing Quaternary palaeo-environments across vast areas of mid- and low-latitudes where relict periglacial assemblages frequently occur, but their full potential remains to be exploited. It is the very first viable solution that seeks to interpret relict periglacial features quantitatively and in a replicable and subjectivity-suppressed manner and, as such, it can provide much more plausible periglacial-based palaeo-air temperature reconstructions than before. Hopefully, it will be a springboard for follow-up developments of more sophisticated modelling tools that will further increase the exploitability and reliability of relict periglacial features as indicators of palaeo-climates.

## Appendix A:   Thermal conductivity of the thawed ground

Thermal conductivity of the thawed ground is calculated as a function of its dry bulk density and volumetric moisture content using the Johansen's (1977) thermal-conductivity model. This empirical transfer scheme requires less parameterizations than its derivatives, but still provides reasonable and consistent estimates of thermal conductivity for a wide range of substrates having the degree of saturation of >5–10 to 100 % (e.g., Dong et al., 2015; He et al., 2017; Zhang et al., 2018). Basically, it interpolates between dry $k_{\mathrm{dry}}$ [W m$^{-1}$ K$^{-1}$] and saturated $k_{\mathrm{sat}}$ [W m$^{-1}$ K$^{-1}$] thermal conductivity of a material as follows:

$$k_{\mathrm{t}} = k_{\mathrm{dry}} + (k_{\mathrm{sat}} - k_{\mathrm{dry}})K_{\mathrm{e}}, \tag{A1}$$

where $K_{\mathrm{e}}$ [$-$] is the dimensionless Kersten number used to normalize the contrast between $k_{\mathrm{sat}}$ and $k_{\mathrm{dry}}$ based on the degree of saturation.

The dry ground thermal conductivity is defined by the following semi-empirical relationship (Johansen, 1977):

$$k_{\mathrm{dry}} = \frac{0.135\rho + 64.7}{2700 - 0.947\rho}, \tag{A2}$$

where the constant of $2\,700\,\mathrm{kg\,m^{-3}}$ represents the typical density of solid ground particles that is, for consistency, used throughout the thermal-conductivity scheme.

The saturated ground thermal conductivity is calculated by a weighted geometric mean based on the thermal conductivities of individual ground constituents and their respective volume fractions (Johansen, 1977):

$$k_{\mathrm{sat}} = k_{\mathrm{s}}^{1-n} k_{\mathrm{w}}^{n}, \tag{A3}$$

where $k_{\mathrm{s}}\,[\mathrm{W\,m^{-1}\,K^{-1}}]$ and $k_{\mathrm{w}}\,[\mathrm{W\,m^{-1}\,K^{-1}}]$ is the thermal conductivity of solid ground particles and water, respectively, and $n\,[-]$ is the ground porosity, which is expressed as a function of the dry bulk density and the typical density of solids:

$$n = 1 - \frac{\rho}{2700}. \tag{A4}$$

The thermal conductivity of water is set at $0.57\,\mathrm{W\,m^{-1}\,K^{-1}}$, while that of solids is computed as:

$$k_{\mathrm{s}} = k_{\mathrm{q}}^{q} k_{\mathrm{o}}^{1-q}, \tag{A5}$$

where $k_{\mathrm{q}}\,[\mathrm{W\,m^{-1}\,K^{-1}}]$ and $k_{\mathrm{o}}\,[\mathrm{W\,m^{-1}\,K^{-1}}]$ is the thermal conductivity of quartz and other minerals, respectively, and $q\,[-]$ is the quartz fraction of the total content of solids. Quartz is assigned the thermal conductivity of $7.7\,\mathrm{W\,m^{-1}\,K^{-1}}$, while for other minerals it is as follows (Johansen, 1977):

$$k_{\mathrm{o}} = \begin{cases} 3, & \text{for } q < 20\,\% \wedge \text{coarse-grained} \\ 2, & \text{otherwise} \end{cases}. \tag{A6}$$

Note that materials having more than $5\,\%$ of clay should be considered as fine-grained, while others should be treated as coarse-grained (sensu Johansen, 1977).

Since the quartz content is usually unknown, it can be estimated as a weighted average of its empirically obtained percentages for clay, silt, and sand fraction, having 0, 15, and $45\,\%$, respectively, which is thought to have an uncertainty $< 30\,\%$ (Johansen, 1977). Alternatively, it can also be drawn from a nomogram using the ground texture (Fig. A1).

The Kersten number for thawed fine- and coarse-grained ground with the degree of saturation $S\,[-]$ larger than $10\,\%$ and $5\,\%$, respectively, is expressed as (Johansen, 1977):

$$K_{\mathrm{e}} = \begin{cases} \log S + 1, & \text{for } S > 10\,\% \wedge \text{fine-grained} \\ 0.7\log S + 1, & \text{for } S > 5\,\% \wedge \text{coarse-grained} \end{cases}, \tag{A7}$$

where

$$S = \frac{\phi}{n}. \tag{A8}$$

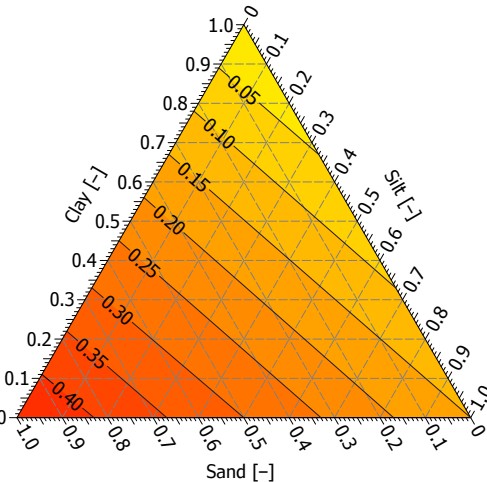

**Figure A1.** A nomogram for estimating the quartz content using the ground texture based on Johansen (1977). Note that the quartz content (black solid diagonal lines) is obtained at the intersection of the contents of clay, silt, and sand (grey dashed three-way lines) that are read in the direction of their respective tick marks and labels.

Clearly, the values of the calculated ground thermal conductivity increase exponentially and logarithmically with rising input values of the dry bulk density and volumetric moisture content, respectively, and thus the conductivity tends to change more sharply if the inputs are higher and lower, respectively (Fig. A2).

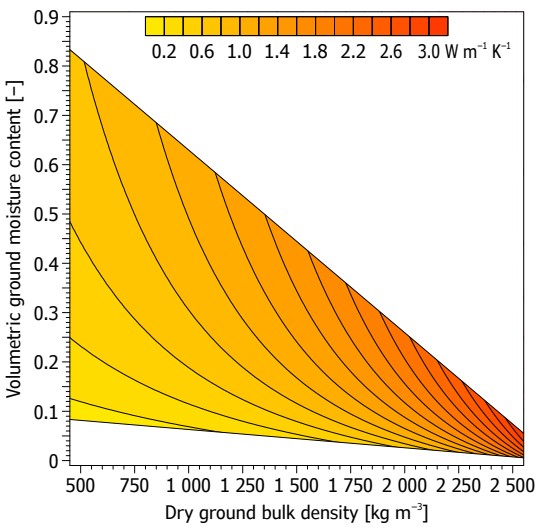

**Figure A2.** Thermal conductivity of the thawed ground as a function of its dry bulk density and volumetric moisture content calculated using the Johansen's (1977) thermal-conductivity model for thawed fine substrates with the degree of saturation of >10 to 100 % and the quartz content of 40 %. Note that the white areas are outside the saturation limits.

**Table C1.** Variables used to model air temperature characteristics at the James Ross Island (upper section) and Alaskan Arctic (lower section) validation sites. Note that the superscripts *norm* and *unif* indicate normal and uniform distributions describing the respective variables.

| Site | $\xi$ [m]$^{norm}$ | $\phi$ [−] | $\rho$ [kg m$^{-3}$] | $q$ [−]$^{unif}$ | $f/c$ | $n_t$ [−]$^{norm}$ | $A_a$ [°C]$^{norm}$ |
|---|---|---|---|---|---|---|---|
| Abernethly Flats | 0.57±0.04 | 0.250 | 1380 | 0.18–0.34 | fine | 2.30±0.55 | 19.7±1.6 |
| Berry Hill slopes | 0.83±0.02 | 0.290 | 1850 | 0.22–0.40 | fine | 2.68±0.44 | 18.6±2.5 |
| Johann Gregor Mendel | 0.55±0.06 | 0.147 | 1460 | 0.25–0.46 | fine | 3.38±0.72 | 20.4±2.2 |
| Johnson Mesa | 0.54±0.06 | 0.197 | 1435 | 0.18–0.46 | fine | 3.05±0.93 | 19.1±2.3 |
| Deadhorse | 0.72±0.04 | 0.515 | 980 | 0.06–0.11 | fine | 0.97±0.18 | 40.2±3.5 |
| Franklin Bluffs | 0.63±0.04 | 0.583 | 1125 | 0.07–0.13 | fine | 0.53±0.09 | 43.7±3.1 |
| West Dock | 0.29±0.06 | 0.725 | 413 | 0.00–0.00 | fine | 0.49±0.06 | 37.3±3.6 |

See Table 1 for abbrevations.

## Appendix B: Solution using MATWM to define the range of annual air temperature oscillations

Unconventionally, the range of annual air temperature oscillations can also be defined using MATWM (cf. Williams, 1975) if its difference from MAAT (i.e., MATWM − MAAT) is substituted for $\frac{A_a}{2}$ in Eq. (4) and elsewhere. Likewise, MAAT is then calculated numerically using the bisection root-finding algorithm, but is searched for $-\infty < \text{MAAT} \leq 0$ because the actual value of the annual air temperature range is to be determined by the calculation itself. This solution can be advantageously combined with other palaeo-indicators that allow to estimate MATWM. However, it should be employed cautiously because it

is highly sensitive to variations of $I_{ta}$ and MATWM itself (Fig. 2), which can result in major errors if the input variables are defined inaccurately. Unfortunately, the deviations are expected to be higher at lower both $I_{ta}$ and MATWM (Fig. 2) that are characteristic for permafrost regions. Note that this alternate solution is also implemented in the PERICLIMv1.0 R package.

## Appendix C: Validation data

The PERICLIMv1.0 validation was based on mostly previously published data obtained in the period 2011/2012 to 2017/
2018 at four bare permafrost sites located on James Ross Island, north-eastern Antarctic Peninsula, between 63°49′–63°53′ S, 57°50′–57°57′ W, and 10–340 m asl (e.g., Hrbáček et al., 2017a, b; Hrbáček and Uxa, 2020), and data collected by the Geophysical Institute Permafrost Laboratory at the University of Alaska Fairbanks in the period 2001/2002 to 2016/2017 at three vegetated permafrost locations on the coastal plain of the Alaskan Arctic adjacent to the Beaufort Sea, between 69°40′–70°22′ N, 148°28′–148°43′ W, and 3–111 m asl (https://permafrost.gi.alaska.edu/sites_list, access: 28 June 2019; Romanovsky
et al., 2009; Wang et al., 2018) (Table C1). The stations measured air and ground temperatures with thermistor sensors installed in solar radiation shields 1.5 or 2 m above ground surface and at six to fifteen depth levels ranging from or near from the ground surface to 0.75 m or around 1 m below, and their records were averaged to daily resolution (Romanovsky et al., 2009; Hrbáček et al., 2017a, b; Wang et al., 2018; Hrbáček and Uxa, 2020).

Active-layer thickness, which corresponds to the maximum annual depth of the 0 °C isotherm (Burn, 1998), was primarily determined by a linear interpolation between the depths of the deepest and the shallowest sensors with the maximum annual ground temperature $> 0\,°C$ and $\leq 0\,°C$, respectively. Alternatively, it was established by a linear extrapolation of the maximum annual ground temperatures of the two deepest sensors if both were positive. Thawing and freezing seasons were defined by a continued prevalence of positive and negative mean daily temperatures, respectively, at the shallowest ground temperature sensor and, for consistency, these time-windows were also utilized for air temperatures. Since the model assumes that air temperatures are solely positive and negative during the thawing and freezing season, respectively (Fig. 1), positive and negative air temperatures alone were taken to determine MATTS, $I_{\text{ta}}$, $L_{\text{t}}$ and MATFS, $I_{\text{fa}}$, $L_{\text{f}}$. Likewise, MAAT was calculated as a length-weighted average of the seasonal air temperature means for a period composed of the thawing season and its preceding freezing season, which is thought to be more representative for the active-layer formation than a fixed calendar period (Hrbáček and Uxa, 2020). Annual air temperature range was defined by an annual spread of a 31-day simple central moving average of mean daily air temperatures, with its extremes being considered to substitute MATWM and MATCM. Finally, ground-surface thawing $n$-factor was derived as a ratio of the thawing index at the shallowest ground temperature sensor and air thawing index. Consequently, for modelling, the active-layer thickness had to be reduced by the depth of the shallowest ground temperature sensor to ensure consistency of the calculations because the model presumes that the ground-surface thawing $n$-factors transfer between air and ground-surface temperatures (cf. Hrbáček and Uxa, 2020).

Ground physical properties for the James Ross Island sites were determined in situ or from intact samples collected near the temperature monitoring stations at a depth of 0.1–0.3 m during the thawing seasons 2013/2014 to 2018/2019 (Hrbáček et al., 2017a; Hrbáček and Uxa, 2020), while those for the Alaskan sites were adapted from Zhang (1993) and Romanovsky and Osterkamp (1997) who took samples from a depth of up to about 0.6 m during the thawing season 1991 and then averaged their characteristics over the full active-layer thickness. Volumetric ground moisture content was established by successive wet and dry weighing or through replicate measurements with time-domain reflectometry probes. Similarly, dry ground bulk density was determined using dry weighing (Zhang, 1993; Hrbáček et al., 2017a; Hrbáček and Uxa, 2020). Ground quartz content was estimated on the basis of the proportions of clay, silt, and sand fractions (see Appendix A), which were ascertained through a wet sieving and X-ray diffraction (Hrbáček et al., 2017a; Hrbáček and Uxa, 2020) or assessed visually via soil type (Zhang, 1993). All the substrates were considered as fine-grained owing to their relatively high clay–silt contents.

**Appendix D: Ground-surface and air thawing index in a two-layer ground**

If two distinct ground layers can be distinguished within the palaeo-active layer, the Stefan equation for calculating the active-layer thickness in two-layer ground can be applied. It has been proposed in the following form (Nixon and McRoberts, 1973; Kurylyk, 2015):

$$\xi = -Z_1 \frac{k_{\text{t}_2}}{k_{\text{t}_1}} + Z_1 + \sqrt{\frac{Z_1^2 k_{\text{t}_2}^2}{k_{\text{t}_1}^2} + \frac{2 k_{\text{t}_2} I_{\text{ts}}}{L \phi_2 \rho_{\text{w}}} - \frac{Z_1^2 k_{\text{t}_2} \phi_1}{k_{\text{t}_1} \phi_2}}, \tag{D1}$$

where $Z_1$ [m] is the thickness of the top sub-layer, the physical properties of which are subscripted by 1, while the bottom sub-layer is denoted by the subscripts 2. The ground-surface index can then be simply expressed from Eq. (D1):

$$I_{\text{ts}} = \frac{\left[ (\xi + Z_1 \frac{k_{t_2}}{k_{t_1}} - Z_1)^2 - \frac{Z_1^2 k_{t_2}^2}{k_{t_1}^2} + \frac{Z_1^2 k_{t_2} \phi_1}{k_{t_1} \phi_2} \right] L \phi_2 \rho_{\text{w}}}{2 k_{t_2}}. \tag{D2}$$

As in Eq. (1 and 2), the product of $\phi$ and $\rho_{\text{w}}$ can be substituted by that of the gravimetric moisture content and dry bulk density of the ground, but note that the fraction on the far right of Eq. (D1) and at the corresponding place of Eq. (D2) is simplified because the density of water in its numerator and denominator is the same. Subsequent procedures to derive the air temperature characteristics are analogous to those for the one-layer solution (Eq. 3 to 14).

*Code and data availability.* The latest version of PERICLIMv1.0 is available as R package from https://github.com/tomasuxa/PERICLIMv1.0 under the GPLv3 license. The exact version of the model used to produce this paper is archived at https://doi.org/10.5281/zenodo.4562435. The validation datasets from James Ross Island are available upon request from FH (hrbacekfilip@gmail.com), whereas those from Alaskan Arctic can be retrieved from https://permafrost.gi.alaska.edu/sites_list.

*Author contributions.* TU came up with an initial idea with feedbacks from MK, developed the model, evaluated it against data from James Ross Island and Alaskan Arctic, which were processed by FH and TU, respectively, and tested it for derivation of palaeo-air temperature characteristics using data sampled collectively with MK in the Czech Republic. TU drew figures and wrote the manuscript with inputs from MK and FH. All authors reviewed and approved the final version of the paper.

*Competing interests.* The authors declare that they have no conflict of interest.

*Acknowledgements.* We thank Tereza Dlabáčková for her assistance at the Nebanice site and with sample analysis. The Geophysical Institute Permafrost Laboratory at the University of Alaska Fairbanks is acknowledged for its continuous effort in collecting temperature data across Alaska and their online dissemination. Last but not least, we thank the reviewers and the editor who contributed significantly to the improvement of the paper.

*Financial support.* The PERICLIMv1.0 development and evaluation was supported by the Czech Science Foundation, project number 17-21612S. The validation datasets from James Ross Island were collected thanks to the Ministry of Education, Youth and Sports of the Czech Republic, project number VAN2020/01.

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
