# Peer review of "PERICLIMv1.0: A model deriving palaeo-air temperatures from thaw depth in past permafrost regions"

_Geoscientific Model Development, 2020_

## Referee Comment (RC1) · Anonymous Referee #1 · 25 Feb 2020

"PERICLIMv1.0: A model deriving palaeo-air temperatures from thaw depth in past permafrost regions", by Tom.š Uxa, Marek Kˇr.žek, and Filip Hrbáˇcek

**General comments**

The discussion paper describes a modelling scheme to inversely estimate the near-surface atmospheric thermal states from the thaw depth information under permafrost conditions, primarily employing the idealized relationship derived from the Stefan formula, and discussed the applicability of the model to infer the past thermal conditions from the relict periglacial features, namely, active layer thickness. The authors demonstrated the efficiency of the scheme to inversely estimate the temperature characteristics, such as mean annual air temperature, and mean air temperatures of coldest and warmest months, from the active layer thickness observed at the Antarctic and Arctic sites.

The relationship expressed as the Stefan solution, Equation (1) in the text, is widely known for its useful simplicity but also tendency for biases when applied to real observations, as partly stated in the discussion paper. Still, the author proposed a new and intriguing idea to apply the relationship as a modelling framework to infer the paleo-thermal conditions at the formation time of the currently relict periglacial feature, which can be very relevant to geoscientific modelling within the scope of GMD, as well as of paleoclimatology and cryosphere-related science. In the current form of the paper, however, the explanation and evaluation of the modelling scheme are confusing, or poorly written in term of model description, and the title and the target of the paper show substantial mismatches with the current structure of the paper. Thus, the reviewer believes that the manner conducting the model evaluation, as well as overall organization of the presented text need substantial revisions (appreciating the effort that the manuscript went overall rearrangements for a model description paper), as well as the way the model application was conducted and presented in the Result section.

Two major suggestions are:

1. Move the subsection 5.4 to the Introduction section to describe the previous studies, or motivation of the study, and reconstruct the whole text to fit to a description paper of a model to be used for paleo-temperature reconstructions.

2. Add a model validation case. Use only those terms and variables that would be available and used in paleo application cases (for example, seting $P$ to 365 days, and $Aa$ as described in 5.2.3, etc.), and see how the computed atmospheric thermal states compare to the observed. Also, sensitivity tests for parameters (eg, physical properties; thawing n-factor) to evaluate the range of variations in the computed temperatures would be very informative addition to discussions in 5.2.

**Specific comments**

Abstract:

The title and abstract claim that this paper introduces a presented model is to be used for paleo-temperature reconstructions. The reviewer feels that the authors' intention and the current structure and the way the model was run and evaluated have a large gap, and suspect that the current evaluation against modern temperature records could not serve properly to judge the model's ability when applied to the past periglacial features. The performed evaluation seemed to use information that could not be available for the paleo cases.

P. 2, ll. 46-48: This comment is related to the above one. If the paper is intended to introduce a model, it should show clear structure of the model, preferably with a simple schematic diagram, to show, for example, what are the input variables to the model; what are the parameters to be set or assumed; and what are the output variables that the model produce. The current paper appears merely to demonstrate a Stefan-based calculation scheme using the observed temperature and relevant data. Thus, if it is intended to evaluate the model to infer air temperature characteristics from past periglacial evidence, it should demonstrate the way the model would perform when applied to paleo cases (See the related comments in the "Result" section). An example of a modelling scheme for paleo application would be like:

[Input] active layer thickness

[Parameters to be determined, assumed, or deduced] thermal conductivity (thawed), wetness, thawing n-factor, length of the period (fixed at 365 days), annual air temperature amplitude

[Outputs] thermal conditions and related information (MAAT, MATW/CM, MATT/FS, Lt, Lf…)

P. 4, l. 84 (and others), "amplitude":
This is merely a suggestion. Use of a word "range" to denote a margin between the lowest and highest values, maybe useful to distinguish $Aa$ and $Aa/2$ in the text.

P. 5, ll. 98-106: Estimation of MAAT and $Aa$ appears the key, or the central part of the model when applied to the paleo settings when no a-priori thermal knowledge is available. Under the current modelling framework, when one assumes a sinusoidal annual temperature change, and has the $Ita$ value as the area under the curve for the positive values, MAAT and $Aa$ can be determined independently. In the current form, it is not clear if $Aa$ is a parameter or an output variable in the model. So, it definitely needs more elaboration to describe how to calculate (or estimate) MAAT and/or $Aa$ (with this in mind that this modelling scheme is to be used for the paleo applications).

P. 5, ll. 105-106, "However, potential drawbacks can be easily handled if exclusively permafrost-related features are examined.": It is not clear what is intended to say.

P. 6, Figure 2. It would be very user-friendly to describe how to draw relevant information from the figure, for most of the GDM reader won't be familiar with this nomogram.

P. 7, Table 2, and "Model validation": Please comment on the applicability to "Berry Hill slopes" in the text (other James Ross sites appear on the flat locations).
Applicability of the Stefan solution may be limited to those sites with high vertical heterogeneity, and it is mentioned in the text for the Alaskan site. Applicability to sites with large lateral flows of heat or water would also be limited.

P. 7, ll. 128-129, "The Stefan equation": Does it mean Eq. (1)? If so, please add the notation (similar to p. 10, l. 212). Also, it is not clear, what are the difference between the results of what this sentence means, and what the later demonstration of the model in "Result" section.

P. 8, ll. 162-163: It is curious if the results from the "successive wet and dry weighing" and the TDR probes are consistent to each other, or were independently done and not comapred (eg, Alaskan sites were solely done by the former, and the James Ross by latter). Is it possible to mention the representative of the results to be applied to the entire thawed layer?

PP. 8-10. Although not clearly written, it seems that the results shown in this section used some of the temperature information obtained from the observations (for example, $P$ and $Aa$ as shown in Table 2, which claims "model-driving" parameters). If the purpose of the paper is to demonstrate the ability to reconstruct the temperature conditions derived solely from the geomorphological evidences (that is, depth of the active layer), the evaluation should be done in the same manner as to be intended for the paleo cases. This means to run the model with the input (depth) and assumed parameters (thermal conductivity, wetness, thawing n-factor) only. Otherwise, it appears just a mere application of the calculation scheme using advantage of the present-day observations.

Subsection "5.1 Model uncertainties,…":
This subsection needs to restructure the organization, and clearly reformulate sentences. There are many long sentences with unclean meaning (for example, ll. 222-224, 230-233, 235-237). Also, the discussion sometimes goes back and forth, right and left, with reservation and euphemism. One suggestion for a re-organization would be to first divide the discussion to "strength of the model" and "weakness of the model", and prioritize the issues under each of the categories.

P. 10, l. 214, "Also, it assumes that the frozen layer is at 0 °C before thaw.": This looks pre-assumed in the derivation: that is, what is at stake is the temperature at the freezing interface between the thawed and frozen layers, which should be 0°C.

P. 11, 225-227: It seemingly depends on whether MAAT or MATWM (or MATTS) is at question (cf. Figure 3).

P. 11, l. 231: It is not clear what is meant by "which"

P. 11, l. 236: Not a clear sentence. Are the sites "far from being saturated" those in James Ross Island, and sites of "two-layer" those in Alaska? What (or which) does "there" actually mean?

"5.2 Driving data": Maybe a source of confusion in reading the discussion paper is that it is not clear what are the input (driving) data, what are the parameters or boundary conditions that set the calculations, and what are the output of the model to be applied for the paleo setting. What is discussed in this subsection is either parameters (ie, ground physical properties, thawing n-factor) or a part of the output (temperature amplitude), otherwise it does not make sense if the target temperature information is a driving data.

P. 13, l. 278, "it is possible to assess the extremes, between which the moisture likely occurred": Not clear what is meant.

P. 14, l. 288, "these correlations": which correlations between what? Please describe clearly.

"5.3 Implications for paleo-temperature reconstructions": It would be suggested to rename the subsection title, something like "Applicability to periglacial features".

P. 15, ll. 329-331, "we hypothesize that their depth probably rather reflects the position of a transient layer where the contact between the active layer and the uppermost permafrost at the time of their formation oscillated": It is not clear what this sentence is meant.

P. 15, l. 331 "the latter": It is not clear what is indicated.

P. 15, l. 347, "random-sampling methods": how the methods work in the context? With no information, it is not possible to judge the adequacy of the methods.

"5.4 Progress over previous attempts": the content of this subsection should be placed as "motivation" or "previous studies" in the Introduction, and the whole paper should be structured to "introduce and evaluate" a model to infer air temperature from the paleo-periglacial feature. It is strongly suggested that the overall organization and structure of the paper should be revised.

The authors gave proper credit to related work and clearly indicate their own new/original contribution. And the number and quality of references appear appropriate.

---

## Referee Comment (RC2) · Anonymous Referee #2 · 2 Mar 2020

PERICLIMv1.0: A model deriving palaeo-air temperatures from thaw depth in past permafrost regions

By Uxa et al.,

Major

The manuscript has a lot of jargon words in the Abstract and the Introduction. This makes it difficult to understand and follow from the beginning. The authors introduce the work from a very general perspective and do not include specific details applicable to the current study. The results section has only one figure with a lot of unnecessary discussion points, which are well-known and well-documented in the previous works. In addition, most if not all the formulas and notations can be found in Nelson and Outcalt

(1987), listed in references. Note, that Nelson and Outcalt (1987) acknowledge the surface processes and do not jump straight to the Stefan's formula. I have a common criticism, which is well understood by authors, and I appreciate their effort in providing a detailed description of all the pros and cons of their model. I think that the length of the discussion should and could be reduced. Clearly, snow depth and organic peat layer are two major factors that will add a lot of bias to thaw depth calculation. Also, using a simple (one layer) formula has its significant limitations. However, for paleo-temperatures, it could be feasible.

I felt that authors are presenting the model as a proof-of-concept showing that this algorithm might work. The fact that it is performed well for homogenous soil is logical and not surprising. In addition, the model has a higher success rate for continuous permafrost regions, with minimum surface vegetation and climate-driven permafrost conditions (Shur et al., 2007). I do not think that the model will work well for the discontinuous permafrost areas. I suggest looking at early works by Clow (1992) on temperature inversion, that captures all the complexity dealing with inverse modeling studies applied to permafrost temperature reconstructions.

My major disappointment is that I was expecting to see how the model derives paleo-air temperatures on specific examples. That will be the best justification for me that high order bias can be neglected for paleo-air reconstruction. I have mixed feelings about this work. I appreciate the authors' effort and think that it can be valuable for a paleo-temperature reconstruction. I would be willing to suggest this work for publication once the authors will revise and paper, improve the flow, and get rid of jargon. Ideally, it would be nice to see some paleo-reconstructions cases. I suggest to be more specific from the beginning and clearly state the goal of this work. Below, I suggested minor improvements.

Minor

L2 Not sure what are the climatic controls? Rephrase and clarify.

L5 Which 'flaws'?

L6 What are the relict permafrost related features?

L11-12. Not sure what do you mean. Be more specific.

L14 'relict permafrost features', need to define them first.

L17-18 'active features', 'relict periglacial assemblages' need to define them as well.

L29 'periglacial features', specify.

L30 'geometric attributes', not sure what do you mean by that?

L34 'dimension of features', specify

L69. Why authors did not Kudryavstsev's formula instead, which incorporates the effect of soil moisture, snow, and vegetation. Need to better explain the choice, why not use more sophisticated numerical models like GIPL or Gryogrid?

Table 1 where thermal conductivities and porosities come from? Adding the effect of the organic layer will change the results of the thaw depth (e.g. Jafarov and Schaefer 2016).

L105 not sure what authors mean. Rephrase and add more clarity.

Table 2 Again specify where thermal conductivities and porosities come from.

L140 not sure why extrapolated ALT was 0.15m. it does not make sense.

L197. I would be super cautious with the high accuracy statements.

The rest of the discussion talks about caveats and explains when and why it fails. It is a fair discussion, but I found it rather long and not necessary. All these things are well-known and I would suggest to reduce it to a short summary of the pros and cons. I would rather see the applications as a justification of that this simple method was developed for a reason.

[Figure]

**References**

Clow, G.D. The extent of temporal smearing in surface-temperature histories derived from borehole temperature measurements. Global and planetary change, 6(2), 81-86 (1992)

Shur, Y. L. and Jorgenson, M. T.: Patterns of permafrost formation and degradation in relation to climate and ecosystems, Permafrost Periglac., 18, 7–19, doi:10.1002/ppp.582, 2007.

Jafarov, E. and Schaefer, K.: (2016), The importance of a surface organic layer in simulating permafrost thermal and carbon dynamics, The Cryosphere, 10, 465-475, doi:10.5194/tc-10-465-2016.
* * *

---

## Referee Comment (RC3) · Anonymous Referee #3 · 9 Mar 2020

I read and totally agree with the comments from reviewers 1 & 2. The main issue is the mismatching between the title and content. From the title, as a reader, I would like to know how we can use active layer thickness to inverse palaeo-air temperature. However, the 'inversion 'model seems not to be able to look backward for past years, decades, or centuries. Based on previous studies about this theme, the climate signal was stored in deep permafrost thermal condition (e.g., Clow (1992), Huang et al., (2000)) while the active layer is only several meters below the ground surface, which is strongly influenced by seasonal variation. The current study is only to calculate the present (rather than palaeo-) mean annual air temperature by using active layer thickness given fixed other parameters. Furthermore, assuming $A_a$ maybe not acceptable for reconstructing palaeo-climate. $P$ should be 365 (or 366) rather than ranges from

300+ to 400+(in Table 2).

Thus, the current version is not able to be published but I would be willing to suggest this work for publication once the authors will show some palaeo-climate reconstructions results using active layer thickness.

**References**

Clow, G. D. (1992). The extent of temporal smearing in surface-temperature histories derived from borehole temperature measurements. Global and Planetary Change, 6(2-4), 81-86.

Huang, S., Pollack, H. N., & Shen, P. Y. (2000). Temperature trends over the past five centuries reconstructed from borehole temperatures. Nature, 403(6771), 756-758.

---

## Editor Comment (EC1) · Andrew Wickert (Editor) · 22 Jun 2020

The authors have made a good-faith effort to respond to all referee comments. In my mind, these center on (1) the mismatch between the title of the paper and the "proof of concept" scope in the authors' minds; (2) a need for overall increased clarity; and (3) a need for additional testing and evaluation. I look forward to seeing a revised manuscript that addresses the referees' careful and constructive contributions.

---

## Author Comment (AC1) · 22 Jun 2020

**Author's response to comments of Referee #1**

**AC:** We thank the anonymous referee for the detailed review of our manuscript.

**RC1:** General comments
The discussion paper describes a modelling scheme to inversely estimate the near-surface atmospheric thermal states from the thaw depth information under permafrost conditions, primarily employing the idealized relationship derived from the Stefan formula, and discussed the applicability of the model to infer the past thermal conditions from the relict periglacial features, namely, active layer thickness. The authors demonstrated the efficiency of the scheme to inversely estimate the temperature characteristics, such as mean annual air temperature, and mean air temperatures of coldest and warmest months, from the active layer thickness observed at the Antarctic and Arctic sites.

The relationship expressed as the Stefan solution, Equation (1) in the text, is widely known for its useful simplicity but also tendency for biases when applied to real observations, as partly stated in the discussion paper. Still, the author proposed a new and intriguing idea to apply the relationship as a modelling framework to infer the paleo-thermal conditions at the formation time of the currently relict periglacial feature, which can be very relevant to geoscientific modelling within the scope of GMD, as well as of paleoclimatology and cryosphere-related science. In the current form of the paper, however, the explanation and evaluation of the modelling scheme are confusing, or poorly written in term of model description, and the title and the target of the paper show substantial mismatches with the current structure of the paper. Thus, the reviewer believes that the manner conducting the model evaluation, as well as overall organization of the presented text need substantial revisions (appreciating the effort that the manuscript went overall rearrangements for a model description paper), as well as the way the model application was conducted and presented in the Result section.

**AC:** We agree with the referee and admit that there is a mismatch between the title and content of the original version of the manuscript. Originally, it was meant to be a proof-of-concept study showing that the model performs well on present-day data, providing its best possible validation, which was to demonstrate that it could also reasonably derive past temperature conditions, but now we recognize that its real application on palaeo-periglacial features is necessary. Consequently, we also intend to include in the revised version of the manuscript a palaeo-air temperature reconstruction using a palaeo-active-layer thickness and to compare its outputs with reconstructions based on other proxy records and/or model products. This will also bring changes of the manuscript structure, which will be done in accordance with the referee's suggestions given here and below.

**RC1:** Two major suggestions are:
1. Move the subsection 5.4 to the Introduction section to describe the previous studies, or motivation of the study, and reconstruct the whole text to fit to a description paper of a model to be used for paleo-temperature reconstructions.

**AC:** Agreed. This subsection will be removed from the discussion and its parts will be incorporated into the introduction of the revised version of the manuscript, which will also be made less general and more related to the aims of the manuscript as suggested.

**RC1:** 2. Add a model validation case. Use only those terms and variables that would be available and used in paleo application cases (for example, seting $P$ to 365 days, and $Aa$ as described in 5.2.3, etc.), and see how the computed atmospheric thermal states compare to the observed. Also, sensitivity tests for parameters (eg, physical properties; thawing n-factor) to evaluate the range of variations in the computed temperatures would be very informative addition to discussions in 5.2.

**AC:** We intend to include in the revised version of the manuscript a palaeo-air temperature reconstruction using a palaeo-active-layer thickness and to compare its outputs with reconstructions based on other proxy records and/or model products. A section containing present-day data will be retained in the revised version of the manuscript in order to provide model validation using analogous data available for the palaeo-cases and to perform sensitivity tests.

**RC1:** Specific comments

Abstract:

The title and abstract claim that this paper introduces a presented model is to be used for paleo-temperature reconstructions. The reviewer feels that the authors' intention and the current structure and the way the model was run and evaluated have a large gap, and suspect that the current evaluation against modern temperature records could not serve properly to judge the model's ability when applied to the past periglacial features. The performed evaluation seemed to use information that could not be available for the paleo cases.

**AC:** As stated above, originally, it was meant to be a proof-of-concept study, but we intend to include in the revised version of the manuscript a palaeo-air temperature reconstruction using a palaeo-active-layer thickness and to compare its outputs with reconstructions based on other proxy records and/or model products. Still, a section containing present-day data will be retained in the revised version of the manuscript in order to provide model validation using analogous data available for the palaeo-cases and to perform sensitivity tests.

**RC1:** P. 2, ll. 46-48: This comment is related to the above one. If the paper is intended to introduce a model, it should show clear structure of the model, preferably with a simple schematic diagram, to show, for example, what are the input variables to the model; what are the parameters to be set or assumed; and what are the output variables that the model produce. The current paper appears merely to demonstrate a Stefan-based calculation scheme using the observed temperature and relevant data. Thus, if it is intended to evaluate the model to infer air temperature characteristics from past periglacial evidence, it should demonstrate the way the model would perform when applied to paleo cases (See the related comments in the "Result" section). An example of a modelling scheme for paleo application would be like:

[Input] active layer thickness

   [Parameters to be determined, assumed, or deduced] thermal conductivity (thawed), wetness, thawing n-factor, length of the period (fixed at 365 days), annual air temperature amplitude

[Outputs] thermal conditions and related information (MAAT, MATW/CM, MATT/FS, Lt, Lf…)

**AC:** Please note that Table 1 in the original version of the manuscript shows what variables are inputs (upper section) and what variables are outputs (lower section) and it is incorporated in the section 2.1 describing driving parameters of the model.

As stated above, originally, it was meant to be a proof-of-concept study, but we intend to include in the revised version of the manuscript a palaeo-air temperature reconstruction using a palaeo-active-layer thickness and to compare its outputs with reconstructions based on other proxy records and/or model products, which should also clearly show how the input parameters should be chosen. Still, a section containing present-day data will be retained in the revised version of the manuscript in order to provide model validation using analogous data available for the palaeo-cases and to perform sensitivity tests.

**RC1:** P. 4, l. 84 (and others), "amplitude":

This is merely a suggestion. Use of a word "range" to denote a margin between the lowest and highest values, maybe useful to distinguish $Aa$ and $Aa/2$ in the text.

**AC:** Agreed. A collocation "annual air temperature range" will be used in the revised version of the manuscript instead of "annual air temperature amplitude".

**RC1:** P. 5, ll. 98-106: Estimation of MAAT and *Aa* appears the key, or the central part of the model when applied to the paleo settings when no a-priori thermal knowledge is available. Under the current modelling framework, when one assumes a sinusoidal annual temperature change, and has the *Ita* value as the area under the curve for the positive values, MAAT and *Aa* can be determined independently. In the current form, it is not clear if *Aa* is a parameter or an output variable in the model. So, it definitely needs more elaboration to describe how to calculate (or estimate) MAAT and/or *Aa* (with this in mind that this modelling scheme is to be used for the paleo applications).

**AC:** Please note that numerous other palaeo-air temperature reconstructions, for instance those based on glacier mass-balance modelling, also frequently utilized present-day climatology (that is, including present-day annual air temperature range) combined with MAAT and/or precipitation perturbations (though these were frequently not listed as model driving parameters) or adjusted annual air temperature range to derive most plausible palaeo-climate scenarios. Consequently, we believe that the presented scheme is meaningful.
Unfortunately, MAAT and $A_a$ cannot be determined independently because $I_{ta}$ depends on both of them, that is, $I_{ta}$ can achieve identical values for various combinations of MAAT and $A_a$. For instance, $I_{ta}$ equals 526 °C d if MAAT and $A_a$ is -4 °C and 20 °C, respectively, but also if it is -8 °C and *ca*. 29.8 °C. Consequently, $A_a$ must be a model parameter in order to estimate MAAT and other air temperature characteristics. However, please note that $A_a$ does not define their precise values. Moreover, it is believed to undergo substantially lower temporal variations than, for instance, MAAT, and thus $A_a$ for palaeo-applications may be approximated by its present-day value at a site with relict periglacial features or it may be adjusted based on other regional proxies. We will stress it more in the revised version of the manuscript.

**RC1:** P. 5, ll. 105-106, "However, potential drawbacks can be easily handled if exclusively permafrost-related features are examined.": It is not clear what is intended to say.

**AC:** Since the solution is designed to be used in permafrost environments, problems may occasionally arise in situations where seasonally frozen ground coexists under negative mean annual air temperature (MAAT) because permafrost–seasonal frost boundary rarely coincides exactly with MAAT of 0 °C. So the model should be applied on those features that indisputably formed in the presence of permafrost. Nonetheless, the statement will be changed in the revised version of the manuscript in order to be more understandable.

**RC1:** P. 6, Figure 2. It would be very user-friendly to describe how to draw relevant information from the figure, for most of the GDM reader won't be familiar with this nomogram.

**AC:** We will extend the caption and briefly describe how to draw information from the figure in the revised version of the manuscript.

**RC1:** P. 7, Table 2, and "Model validation": Please comment on the applicability to "Berry Hill slopes" in the text (other James Ross sites appear on the flat locations).
Applicability of the Stefan solution may be limited to those sites with high vertical heterogeneity, and it is mentioned in the text for the Alaskan site. Applicability to sites with large lateral flows of heat or water would also be limited.

**AC:** First of all, it should be noted that for consistency we stick to long-established site names on James Ross Island. Admittedly, the name "Berry Hill slopes" may be a little misleading because the site actually occurs on a

slightly-inclined surface in the foothills of the "Berry Hill", which has a gradient of 5–10° (Hrbáček, F., Nývlt, D., Láska, K.: Active layer thermal dynamics at two lithologically different sites on James Ross Island, Eastern Antarctic Peninsula, Catena, 149, 592–602, http://dx.doi.org/10.1016/j.catena.2016.06.020, 2017), and thus lateral flows of heat or water are supposed to be limited or small. We will mention it in the revised version of the manuscript.

We agree that the applicability of the Stefan solution may be limited at sites having high vertical heterogeneity and those experiencing a certain degree of lateral flows of water or heat. However, we believe it is necessary to test the model in various settings, including those that does not perfectly meet its assumptions, and evaluate its actual performance there. Actually, the minority of locations where the Stefan solution has been applied to estimate the thickness of the active layer can be considered as "ideal".

**RC1:** P. 7, ll. 128-129, "The Stefan equation": Does it mean Eq. (1)? If so, please add the notation (similar to p. 10, l. 212). Also, it is not clear, what are the difference between the results of what this sentence means, and what the later demonstration of the model in "Result" section.

**AC:** Yes, it means Eq. (1). The corresponding notations will be added in the revised version of the manuscript.
The sentence was to state that earlier thaw-depth estimates using the Stefan solution (not estimates of air temperatures based on active-layer thickness) were among the most accurate ever on James Ross Island, while those from the Alaskan sites were among the worst, and just for this reason we believe that these locations are well suited to evaluate the model performance. It will be changed in the revised version of the manuscript in order to be more understandable.

**RC1:** P. 8, ll. 162-163: It is curious if the results from the "successive wet and dry weighing" and the TDR probes are consistent to each other, or were independently done and not comapred (eg, Alaskan sites were solely done by the former, and the James Ross by latter). Is it possible to mention the representative of the results to be applied to the entire thawed layer?

**AC:** Only one of these methods was employed at each study site. Alaskan sites were solely done by successive wet and dry weighing, which was also applied at Abernethy Flats and Johann Gregor Mendel sites on James Ross Island, while the rest of the locations, that is, Berry Hill slopes and Johnson Mesa, was done by a calibrated TDR probe. We will consider using solely the outputs of successive wet and dry weighing in the revised version of the manuscript because it is now available from all the sites on James Ross Island.
Please note that the representativeness of the measurements was already documented by earlier publications cited in the original version of the manuscript (Zhang, 1993; Romanovsky and Osterkamp, 1997; Hrbáček et al., 2017a; Hrbáček and Uxa, 2020), which estimated active-layer thickness there.

**RC1:** PP. 8-10. Although not clearly written, it seems that the results shown in this section used some of the temperature information obtained from the observations (for example, $P$ and $Aa$ as shown in Table 2, which claims "model-driving" parameters). If the purpose of the paper is to demonstrate the ability to reconstruct the temperature conditions derived solely from the geomorphological evidences (that is, depth of the active layer), the evaluation should be done in the same manner as to be intended for the paleo cases. This means to run the model with the input (depth) and assumed parameters (thermal conductivity, wetness, thawing n-factor) only. Otherwise, it appears just a mere application of the calculation scheme using advantage of the present-day observations.

**AC:** As stated above, originally, it was meant to be a proof-of-concept study, but we intend to include in the revised version of the manuscript a palaeo-air temperature reconstruction using a palaeo-active-layer thickness and to compare its outputs with reconstructions based on other proxy records and/or model products, which should also clearly show how the input parameters should be chosen. The model validation based on present-day data will be

done in an analogous manner.

Please note that numerous other palaeo-air temperature reconstructions, for instance those based on glacier mass-balance modelling, also frequently utilized present-day climatology (that is, including present-day annual air temperature range) combined with MAAT and/or precipitation perturbations (though these were frequently not listed as model driving parameters) or adjusted annual air temperature range to derive most plausible palaeo-climate scenarios. Consequently, we believe that the presented scheme is meaningful.

**RC1:** Subsection "5.1 Model uncertainties,…":

This subsection needs to restructure the organization, and clearly reformulate sentences. There are many long sentences with unclean meaning (for example, ll. 222-224, 230-233, 235-237). Also, the discussion sometimes goes back and forth, right and left, with reservation and euphemism. One suggestion for a re-organization would be to first divide the discussion to "strength of the model" and "weakness of the model", and prioritize the issues under each of the categories.

**AC:** The discussion section in the revised version of the manuscript will be reorganized as suggested and sentences will be reformulated to make them clearer.

**RC1:** P. 10, l. 214, "Also, it assumes that the frozen layer is at 0 $^\mathrm{o}$C before thaw.": This looks pre-assumed in the derivation: that is, what is at stake is the temperature at the freezing interface between the thawed and frozen layers, which should be 0 °C.

**AC:** Please note that the Stefan equation does not consider heat conduction below the freeze-thaw plane and as such assumes that temperature is uniformly at 0 °C in the frozen zone (see e.g. Romanovsky, V.E., Osterkamp, T.E.: Thawing of the active layer on the coastal plain of the Alaskan Arctic, Permafrost and Periglacial Processes, 8, 1–22, https://doi.org/10.1002/(SICI)1099-1530(199701)8:1<1::AID-PPP243>3.0.CO;2-U, 1997; Kurylyk, B.L.: Discussion of 'A simple thaw-freeze algorithm for a multi-layered soil using the Stefan equation'by Xie and Gough (2013). Permafrost and Periglacial Processes, 26, 200–206, https://doi.org/10.1002/ppp.1834, 2015). Nonetheless, it will be changed in the revised version of the manuscript in order to be more understandable.

**RC1:** P. 11, 225-227: It seemingly depends on whether MAAT or MATWM (or MATTS) is at question (cf. Figure 3).

**AC:** Agreed. It will be changed in the revised version of the manuscript in order to be more understandable.

**RC1:** P. 11, l. 231: It is not clear what is meant by "which"

**AC:** It should relate to "the Stefan equation tends to deviate", but now we see that it can also ambiguously relate to "the peat-layer thickness in the active layer". It will be changed in the revised version of the manuscript in order to be more understandable.

**RC1:** P. 11, l. 236: Not a clear sentence. Are the sites "far from being saturated" those in James Ross Island, and sites of "two-layer" those in Alaska? What (or which) does "there" actually mean?

**AC:** Rather, it was supposed to be a general statement that the model performed well on present-day data even though some of the validation sites do not perfectly meet the assumptions for the application of the Stefan formula. It will be changed in the revised version of the manuscript in order to be more understandable.

**RC1:** "5.2 Driving data": Maybe a source of confusion in reading the discussion paper is that it is not clear what

are the input (driving) data, what are the parameters or boundary conditions that set the calculations, and what are the output of the model to be applied for the paleo setting. What is discussed in this subsection is either parameters (ie, ground physical properties, thawing n-factor) or a part of the output (temperature amplitude), otherwise it does not make sense if the target temperature information is a driving data.

AC: Table 1 in the original version of the manuscript shows what variables are inputs (upper section) and what variables are outputs (lower section). Values of the input parameters should be obtained directly from relict periglacial structures, while those that cannot be obtained directly should be derived using empirical relations (transfer functions) or should rely on representative published data that allow a meaningful range of their values to be defined. Since we also intend to include in the revised version of the manuscript a palaeo-air temperature reconstruction, it should clearly show the above procedure.

Please note that numerous other palaeo-air temperature reconstructions, for instance those based on glacier mass-balance modelling, also frequently utilized present-day climatology (that is, including present-day annual air temperature range) combined with MAAT and/or precipitation perturbations (though these were frequently not listed as model driving parameters) or adjusted annual air temperature range to derive most plausible palaeo-climate scenarios. Consequently, we believe that the presented scheme is meaningful.

RC1: P. 13, l. 278, "it is possible to assess the extremes, between which the moisture likely occurred": Not clear what is meant.

AC: This should mean that it is possible to determine the range of values, which moisture could reach at the time when periglacial features formed. It will be changed in the revised version of the manuscript in order to be more understandable.

RC1: P. 14, l. 288, "these correlations": which correlations between what? Please describe clearly.

AC: This should mean correlations between thermal conductivity and other ground physical properties. It will be changed in the revised version of the manuscript in order to be more understandable.

RC1: "5.3 Implications for paleo-temperature reconstructions": It would be suggested to rename the subsection title, something like "Applicability to periglacial features".

AC: The subsection name will be changed or the whole subsection will be included into another one in the revised version of the manuscript, given that a palaeo-air temperature reconstruction will be added.

RC1: P. 15, ll. 329-331, "we hypothesize that their depth probably rather reflects the position of a transient layer where the contact between the active layer and the uppermost permafrost at the time of their formation oscillated": It is not clear what this sentence is meant.

AC: Transient layer is a transition zone that alternates in status between seasonally frozen ground and permafrost over sub-decadal to centennial time scales because of natural climate variability (Shur, Y., Hinkel, K.M., Nelson, F.E.: The Transient Layer: Implications for Geocryology and Climate-Change Science, Permafrost and Periglacial Processes, 16, 5–17, https://doi.org/10.1002/ppp.518, 2005), and this is what past periglacial features likely attest to. It will be changed in the revised version of the manuscript in order to be more understandable.

RC1: P. 15, l. 331 "the latter": It is not clear what is indicated.

AC: "the latter" was to be related to the transient layer mentioned in the previous sentence. It will be changed in

the revised version of the manuscript in order to be more understandable.

**RC1:** P. 15, l. 347, "random-sampling methods": how the methods work in the context? With no information, it is not possible to judge the adequacy of the methods.

**AC:** Agreed. Random sampling will be used to generate representative sets of input parameters for a palaeo-air temperature reconstruction, which will be added into the revised version of the manuscript.

**RC1:** "5.4 Progress over previous attempts": the content of this subsection should be placed as "motivation" or "previous studies" in the Introduction, and the whole paper should be structured to "introduce and evaluate" a model to infer air temperature from the paleo-periglacial feature. It is strongly suggested that the overall organization and structure of the paper should be revised.

**AC:** This subsection will be removed from the discussion and its parts will be incorporated into the introduction of the revised version of the manuscript.

**RC1:** The authors gave proper credit to related work and clearly indicate their own new/original contribution. And the number and quality of references appear appropriate.

---

## Author Comment (AC2) · 22 Jun 2020

**Author's response to comments of Referee #2**

AC: We thank the anonymous referee for the review of our manuscript.

**RC2: Major**

The manuscript has a lot of jargon words in the Abstract and the Introduction. This makes it difficult to understand and follow from the beginning. The authors introduce the work from a very general perspective and do not include specific details applicable to the current study. The results section has only one figure with a lot of unnecessary discussion points, which are well-known and well-documented in the previous works. In addition, most if not all the formulas and notations can be found in Nelson and Outcalt (1987), listed in references. Note, that Nelson and Outcalt (1987) acknowledge the surface processes and do not jump straight to the Stefan's formula. I have a common criticism, which is well understood by authors, and I appreciate their effort in providing a detailed description of all the pros and cons of their model. I think that the length of the discussion should and could be reduced. Clearly, snow depth and organic peat layer are two major factors that will add a lot of bias to thaw depth calculation. Also, using a simple (one layer) formula has its significant limitations. However, for paleo-temperatures, it could be feasible.

I felt that authors are presenting the model as a proof-of-concept showing that this algorithm might work. The fact that it is performed well for homogenous soil is logical and not surprising. In addition, the model has a higher success rate for continuous permafrost regions, with minimum surface vegetation and climate-driven permafrost conditions (Shur et al., 2007). I do not think that the model will work well for the discontinuous permafrost areas. I suggest looking at early works by Clow (1992) on temperature inversion, that captures all the complexity dealing with inverse modeling studies applied to permafrost temperature reconstructions.

My major disappointment is that I was expecting to see how the model derives paleo-air temperatures on specific examples. That will be the best justification for me that high order bias can be neglected for paleo-air reconstruction. I have mixed feelings about this work. I appreciate the authors' effort and think that it can be valuable for a paleo- temperature reconstruction. I would be willing to suggest this work for publication once the authors will revise and paper, improve the flow, and get rid of jargon. Ideally, it would be nice to see some paleo-reconstructions cases. I suggest to be more specific from the beginning and clearly state the goal of this work.

AC: We will try to keep the number of jargon words to a minimum in the abstract and introduction in order to be more understandable, but please note that the model is intended to be used mainly by periglacial geomorphologists working in past permafrost environments, and thus some terminology may be difficult to leave. Also, we will make the introduction less general and more related to the aims of the manuscript.

Please note that we do not hide at all that some of the formulas can be already found in Nelson and Outcalt (1987), but we use them and have arranged them differently. Nelson and Outcalt (1987) introduced a scheme that is designed to decide whether permafrost is present at a given location based on air thawing and freezing index, while we seek to derive air temperature conditions using the thickness of the active layer. Definitely, snow cover and organic layer are important factors that affect the thickness of the active layer. Nonetheless, the effect of snow cover does not need to be accounted for in this case because the Stefan equation combined with thawing $n$-factor retrieves air thawing index responsible for a given thickness of the active layer, which is subsequently turned into annual as well as winter air temperature characteristics that are not affected by snow at all. Since most periglacial features develop under bare to grassy surfaces, we believe that the thawing $n$-factor, parameterizing the ground-surface–air temperature relations during the thawing season, can be reasonably estimated based on published values for analogous ground-surface covers. On the other hand, it would be a pure guess in the case of snow cover

thickness and associated freezing *n*-factor. Consequently, we believe that the presented scheme, relying only on the thaw-season ground temperature conditions, is advantageous, also from that point of view that most active-layer features largely develop during the warm part of the year when snow cover is absent or very thin. As for organic layer, please note that it represents a substantial part of the Alaskan profiles included in the original version of the manuscript and indeed causes larger model scatters around the identity lines at these locations (see Figure 3 in the original version of the manuscript), but the overall accuracy is still very good if representative inputs are used, even in the case of an one-layer solution used. We will emphasize the above points more in the revised version of the manuscript and we will also trim down the discussion section.

We consider the referee's statements about the expected model failure in discontinuous permafrost areas as speculative because theoretical studies contrastingly suggested that it should fail rather in very cold locations where permafrost is supposed to be continuous (see Romanovsky, V.E., Osterkamp, T.E.: Thawing of the active layer on the coastal plain of the Alaskan Arctic, Permafrost and Periglacial Processes, 8, 1–22, https://doi.org/10.1002/(SICI)1099-1530(199701)8:1<1::AID-PPP243>3.0.CO;2-U, 1997). If the remark was to be based on the concern that in discontinuous permafrost regions the model could be applied to places with seasonally frozen ground, then we must assure you that this situation should not happen as the model should be exclusively applied on landforms and sedimentary structures indicative of the base of the palaeo-active layer, which indisputably formed in the presence of permafrost mostly during Quaternary cold stages.

We confirm that the original version of the manuscript was meant to be a proof-of-concept study, but now we recognize that real model application on palaeo-periglacial features is necessary. Consequently, we also intend to include in the revised version of the manuscript a palaeo-air temperature reconstruction using a palaeo-active-layer thickness and to compare its outputs with reconstructions based on other proxy records and/or model products.

**RC2: Minor**

L2 Not sure what are the climatic controls? Rephrase and clarify.

**AC:** It should mean the range of climatic conditions, under which individual periglacial features form. It will be changed in the revised version of the manuscript in order to be more understandable.

**RC2:** L5 Which 'flaws'?

**AC:** It is related to the still poorly understood range of climatic conditions, under which individual periglacial features form. It will be changed in the revised version of the manuscript in order to be more understandable.

**RC2:** L6 What are the relict permafrost related features? L11-12. Not sure what do you mean. Be more specific.

**AC:** It means relict (inactive under present-day climate conditions) landforms and sedimentary structures, which formed in the presence of permafrost mostly during Quaternary cold stages. We will try to be more specific in the revised version of the manuscript.

**RC2:** L14 'relict permafrost features', need to define them first.

**AC:** As stated above, it means relict (inactive under present-day climate conditions) landforms and sedimentary structures, which formed in the presence of permafrost mostly during Quaternary cold stages. We will try to be more specific in the revised version of the manuscript.

**RC2:** L17-18 'active features', 'relict periglacial assemblages' need to define them as well. L29 'periglacial features', specify.

**AC:** The sentence will be rephrased in the revised version of the manuscript in order to be more understandable.

**RC2:** L30 'geometric attributes', not sure what do you mean by that? L34 'dimension of features', specify

**AC:** It means morphology (that is, shape and size) of periglacial features. It will be changed in the revised version of the manuscript in order to be more understandable.

**RC2:** L69. Why authors did not Kudryavstsev's formula instead, which incorporates the effect of soil moisture, snow, and vegetation. Need to better explain the choice, why not use more sophisticated numerical models like GIPL or Gryogrid?

**AC:** We build on the Stefan formula because of its simplicity and reasonable accuracy at the same time. Note that the Stefan formula also incorporates soil moisture (see Eq. 1), and the effect of vegetation (~ground-surface cover) is expressed via the empirical thawing $n$-factor (see Eq. 3), which converts ground-surface thawing index into air thawing index. Advantageously, the effect of snow cover does not need to be accounted for because the solution retrieves air thawing index, which is subsequently turned into annual as well as winter air temperature characteristics that are not affected by snow at all. This is advantageous also from that point of view that most active-layer features largely develop during the warm part of the year when snow cover is absent or very thin.

The Kudryavtsev formula requires much more additional inputs as compared to the Stefan formula, such as thawed volumetric heat capacity, frozen thermal conductivity, or mean annual ground temperature at the top of the permafrost, only to derive ground surface temperatures. Numerous other extra inputs related to snow and vegetation, such as their height or thermal conductivity, are required for the conversion between ground-surface and air temperatures (this is done using only the thawing $n$-factor in our solution). Such complexity can admittedly yield better results in present-day applications where the inputs may be easily available, but a larger number of input parameters is unsuitable for palaeo-applications as more numerous assumptions would have to be made. Obviously, it is also the case of the other models, such as GIPL or GryoGrid, which are even more sophisticated and solved numerically.
Some of the above explanations will be incorporated in the revised version of the manuscript.

**RC2:** Table 1 where thermal conductivities and porosities come from? Adding the effect of the organic layer will change the results of the thaw depth (e.g. Jafarov and Schaefer 2016).

**AC:** Table 1 in the original version of the manuscript shows what variables are inputs (upper section) and what variables are outputs (lower section). Values of the input parameters should be obtained directly from relict periglacial structures, while those that cannot be obtained directly should be derived using empirical relations (transfer functions) or should rely on representative published data that allow a meaningful range of their values to be defined. Please note that we intend to include in the revised version of the manuscript a palaeo-air temperature reconstruction, which should clearly show the above procedure.
Organic layer is certainly an issue in active-layer thickness modelling and indeed Figure 5 in the original version of the manuscript nicely documents its effect. Please note that organic layer represents a substantial part of the Alaskan profiles included in the original version of the manuscript and causes larger model scatters around the identity lines at these locations (see Figure 3 in the original version of the manuscript), but the overall accuracy is still very good if representative inputs are used.

**RC2:** L105 not sure what authors mean. Rephrase and add more clarity.

**AC:** Since the solution is designed to be used in permafrost environments, problems may occasionally arise in situations where seasonally frozen ground coexists under negative mean annual air temperature (MAAT) because permafrost–seasonal frost boundary rarely coincides exactly with MAAT of 0 °C. So the model should be applied on those features that indisputably formed in the presence of permafrost. Nonetheless, the statement will be changed in the revised version of the manuscript in order to be more understandable.

**RC2:** Table 2 Again specify where thermal conductivities and porosities come from. L140 not sure why extrapolated ALT was 0.15m. it does not make sense.

**AC:** Values of the input parameters should be obtained directly from relict periglacial structures, while those that cannot be obtained directly should be derived using empirical relations (transfer functions) or should rely on representative published data that allow a meaningful range of their values to be defined. Please note that we intend to include in the revised version of the manuscript a palaeo-air temperature reconstruction, which should clearly show the above procedure.

Please note that L140 does not state that the extrapolated active-layer thickness was at most only 0.15 m, but rather that it was at most 0.15 m below the deepest ground temperature sensor available for the estimation (extrapolation) of the active-layer thickness. Since the sensor was at a depth of 0.75 m, the maximum extrapolated active-layer thickness was 0.90 m. Given the maximum vertical distance between the sensor and extrapolated active-layer thickness is as low as of 0.15 m, it is assumed that the active-layer thickness is plausible and can be used for validation in this manuscript. We feel it is necessary to assure the reader about that because active-layer thickness is sometimes extrapolated to depths well under the deepest ground temperature sensors by other papers and the resulting values may thus be of questionable validity. Nonetheless, we will slightly modify this part in the revised version of the manuscript in order to be more understandable.

**RC2:** L197. I would be super cautious with the high accuracy statements.
The rest of the discussion talks about caveats and explains when and why it fails. It is a fair discussion, but I found it rather long and not necessary. All these things are well-known and I would suggest to reduce it to a short summary of the pros and cons. I would rather see the applications as a justification of that this simple method was developed for a reason.

**AC:** We will moderate our accuracy statements and trim down the discussion in the revised version of the manuscript. Also, a palaeo-air temperature reconstruction will be added.

**RC2:** References

Clow, G.D. The extent of temporal smearing in surface-temperature histories derived from borehole temperature measurements. Global and planetary change, 6(2), 81-86 (1992)

Shur, Y. L. and Jorgenson, M. T.: Patterns of permafrost formation and degradation in relation to climate and ecosystems, Permafrost Periglac., 18, 7–19, doi:10.1002/ppp.582, 2007.

Jafarov, E. and Schaefer, K.: (2016), The importance of a surface organic layer in simulating permafrost thermal and carbon dynamics, The Cryosphere, 10, 465-475, doi:10.5194/tc-10-465-2016.

---

## Author Comment (AC3) · 22 Jun 2020

**Author's response to comments of Referee #3**

**AC:** We thank the anonymous referee for the review of our manuscript.

**RC3:** I read and totally agree with the comments from reviewers 1 & 2. The main issue is the mismatching between the title and content. From the title, as a reader, I would like to know how we can use active layer thickness to inverse palaeo-air temperature. However, the 'inversion 'model seems not to be able to look backward for past years, decades, or centuries. Based on previous studies about this theme, the climate signal was stored in deep permafrost thermal condition (e.g., Clow (1992), Huang et al., (2000)) while the active layer is only several meters below the ground surface, which is strongly influenced by seasonal variation. The current study is only to calculate the present (rather than palaeo-) mean annual air temperature by using active layer thickness given fixed other parameters. Furthermore, assuming Aa maybe not acceptable for reconstructing palaeo-climate. P should be 365 (or 366) rather than ranges from 300+ to 400+ (in Table 2).

Thus, the current version is not able to be published but I would be willing to suggest this work for publication once the authors will show some palaeo-climate reconstructions results using active layer thickness.

References

Clow, G. D. (1992). The extent of temporal smearing in surface-temperature histories derived from borehole temperature measurements. Global and Planetary Change, 6(2- 4), 81-86.

Huang, S., Pollack, H. N., & Shen, P. Y. (2000). Temperature trends over the past five centuries reconstructed from borehole temperatures. Nature, 403(6771), 756-758.

**AC:** We agree with the referee and admit that there is a mismatch between the title and content of the original version of the manuscript. Our original intention was to show that the model performs well on present-day data, providing its best possible validation, which was to demonstrate that it could also reasonably derive past temperature conditions, but now we recognize that its real application on palaeo-periglacial features is necessary. Consequently, we intend to include in the revised version of the manuscript a palaeo-air temperature reconstruction using a palaeo-active-layer thickness and to compare its outputs with reconstructions based on other proxy records and/or model products. Still, a section containing present-day data will be retained in the revised version of the manuscript in order to provide model validation and perform sensitivity tests. Please note that the model is supposed to rely on relict permafrost features, which have formed within the active layer, and as such these features can indicate its former thickness (mostly through characteristic structures found in sedimentary profiles) at locations where permafrost occurred during Quaternary cold stages. It does not exploit temperatures measured in deep boreholes.

---

## Referee Report (RR1)

"PERICLIMv1.0: A model deriving palaeo-air temperatures from thaw depth in past permafrost regions" by Tomáš Uxa, Marek Křížek, and Filip Hrbáček.

This manuscript introduces and demonstrates the validity and sensitivity of a new simple inverse modelling scheme based on Stefan's equation to calculate palaeo-air temperature characteristics from the palaeo-active-layer thickness observed in the past permafrost regions. The revised manuscript has substantially been improved from the original, responding in a sufficient manner to the comments and suggestions raised by the referees. In my opinion, the manuscript is close to acceptance for the publication at the GMD journal after the following issues are adequately addressed and clarified.

Major concerns:

1.  "Driving parameters"
It would be strongly suggested to avoid usage of the inclusive term "driving parameters". From a numerical modelling point of view, the variables collectively categorized as "input" in Table 1 or mentioned in the text are categorized into qualitatively different groups, i.e., those to set up a model, those to control the model, and those fed to the model to produce an output [although it may appear just one group from a field scientists' perspective as they are measured and observed at the same time at the field sites, except for "annual air temperature range")]. Nevertheless, lack of clear distinction between these groups appears to lead to confusions in the analysis and/or interpretations in Sections 4 and 5.
Of the variables listed as "input" in Table 1, moisture content to thawing n-factor provide the site-specific information in this model's framework, and actually determine the condition of the (under)ground at which a periglacial feature occurs. They deserve to be called "parameters" as they determine the shape and functionality of the model. To the contrary, active layer thickness is a result of action that occurred at such a place as set by the above "parameters" under a certain climate condition (i.e., thermal, in this model's case), which is the targeted output of the model. Thus, it works as the "input" or driving term of the model.
Temperature range may be called a controlling parameter to the MAAT as the two of them cannot be determined uniquely by $I_{ta}$ alone (this is also relevant to the next issue).

2.  Functionality of "annual air temperature range" $A_a$
When deriving MAAT from Eq (5 to 7) under the given value of $I_{ta}$, it is trivial that MAAT decreases as $A_a$ increases (leading to increase in the absolute value of $I_{fa}$, decrease in MATCM and MATTS, and increases in $L_f$), in which the value of $A_a$ directly controls the output. From this point of view, the arguments in Section 5 (namely, ll. 367–372) look off the mark. In contrast, the modelled MATCM

(or other variables related to the freezing or cold season) can be used to evaluate the plausible value of $A_a$. For example, the argument shown in section 5.2 (ll. 348–350) could be reversed to discuss possible inference on the annual air temperature range that best explains the value of MATCM (or similar variables) derived from other proxies (e.g., -27 to -16.5℃ in case of Central European lowland).

3. Evaluations on the empirical reconstruction methods.

(This is more or less a diplomatic suggestion.) In Abstract and Introduction section, the authors state that their model is aimed to overcome the "flaws" of the empirical methods which are "far from reliable". Yet, the evaluation of the model performance did rely on the outcomes from the empirical methods (ll. 11–12, Section 5.2). Although an assertion of novelty and superiority of the new method is understandable, it doesn't appear fair. The spirit of the new model should lie in its capability to provide more verifiable reconstructions "in a replicable and subjectivity-suppressed manner" (l. 419).

Minor issues/technical issues:

ll. 19–20, "Commonly,… of past environmental conditions": any reference to support the sentence?

l. 102: "the number of inputs" should be small.

l. 121: Should be Eq (5 to 7) to include boundary conditions.

Table 2: It would be good to provide the number of samples (or sampling points).

l. 230, "supposed to be representative for former conditions as such": not clear. Meant something like "supposed to be unchanged from the time of cryoturbation"?

Section 5.1: It should be mentioned in the preamble that this section considers the results of Section 3 (present-day application).

l. 294, l. 416: How is the "success rate" defined and evaluated?

ll. 311–314, ll. 362–365, ll. 412–415: Sentences are too long, and not clear.

Section 5.2: It should be mentioned in the preamble that this section considers the results of Section 4.3 (palaeo application).

ll. 357–360: Additional evidence or arguments would be required to support or substantiate the claim that not the model outputs but the empirical MAAT thresholds are to be revised.

Section 5.4: It would be suggested to modify the title, for example, "Limitations and applicability of the model".

l. 381: "However, it can also be easily adapted for seasonal-frost features": It won't be that "easily". Basically, adaptation will be a mirror image (e.g., changing the suffix $t$ to $f$), but the estimation and validation of snow conditions (or freezing n-factor) can still be complicated.

l. 383, "involving natural climate as well as active-layer thickness variations": Suggested to revise, e.g., "involving natural variations in climate as well as in active-layer thickness"?

ll. 388–390, "some periglacial features,… microstructures": "small-scale periglacial features" would suffice.

l. 398: What does "co-occurring periglacial features" mean? Periglacial features occurring side-by-side?

l. 401: "a more complete" to "an abundant"?

---

## Referee Report (RR2)

[revised manuscript text omitted]

115 with

$$t_1 = \arcsin\left(-\frac{MAAT}{\frac{A_a}{2}}\right)\frac{P}{2\pi}, \tag{6}$$

$$t_2 = \left[\pi - \arcsin\left(-\frac{MAAT}{\frac{A_a}{2}}\right)\right]\frac{P}{2\pi}, \tag{7}$$

where $t_1$ is the time when the air temperature curve crosses the zero-degree Celsius level from below ($\sim$thawing season begins),
120 while $t_2$ is the time when it crosses this level from above ($\sim$thawing season ends) (e.g., Nelson and Outcalt, 1987).

Unfortunately, Eq. (5) has no analytical solution for MAAT. The latter can be derived from a nomogram (Fig. 2), but here it is calculated numerically using the bisection root-finding method searching for MAAT such that $-A_a < MAAT \leq 0$. This condition ensures that both positive and negative air temperatures have occurred during the year, which is an essential prerequisite for the active layer to form. Admittedly, it is simplistic because air–ground temperatures are modulated by surface and sub-
125 surface offsets so that permafrost–seasonal frost boundary usually occurs at slightly negative MAAT (Smith and Riseborough, 2002). Consequently, there might be a risk that the model is incorrectly applied to seasonal-frost conditions. However, this can be easily prevented if periglacial features that have indisputably developed in the presence of permafrost are examined.

Once MAAT is known, the air freezing index $I_{fa}$ ($^\circ$C d) can be simply computed as:

$$I_{fa} = MAAT\,P - I_{ta}. \tag{8}$$

130 Furthermore, MATWM and MATCM calculated as:

$$MATWM = MAAT + \frac{A_a}{2}, \tag{9}$$

[revised manuscript text omitted]

---

## Author Response (AR2)

**Author's responses**

Dear Dr. Wickert,

On behalf of the authors, I am pleased to submit the second revision of our manuscript ID gmd-2020-5 entitled "PERICLIMv1.0: A model deriving palaeo-air temperatures from thaw depth in past permafrost regions" compiled by Tomáš Uxa, Marek Křížek, and Filip Hrbáček.

We responded point-by-point to all referee #1 comments and made corresponding adjustments in the manuscript. The referee #2 provided comments and suggestions inserted directly in the manuscript. We also followed most of them, which can be seen in the revised manuscript with tracked changes. The reasons why we did not take some of them into account in the text are outlined below for these specific comments and suggestions. Finally, we made several other minor modifications in the manuscript, which you will find using the tracked changes.

All the co-authors have read and approved the manuscript prior to its resubmission to Geoscientific Model Development. No part of the manuscript can infringe upon existing copyrights in any way. We have no conflict of interest to declare.

Thank you very much for reviewing the revised manuscript.

Yours sincerely,
Tomáš Uxa

**Author's response to comments of Referee #1**

AC: We thank the referee #1 for the detailed of our manuscript.

RC1: This manuscript introduces and demonstrates the validity and sensitivity of a new simple inverse modelling scheme based on Stefan's equation to calculate palaeo-air temperature characteristics from the palaeo-active-layer thickness observed in the past permafrost regions. The revised manuscript has substantially been improved from the original, responding in a sufficient manner to the comments and suggestions raised by the referees. In my opinion, the manuscript is close to acceptance for the publication at the GMD journal after the following issues are adequately addressed and clarified.

Major concerns:

1. "Driving parameters"

It would be strongly suggested to avoid usage of the inclusive term "driving parameters". From a numerical modelling point of view, the variables collectively categorized as "input" in Table 1 or mentioned in the text are categorized into qualitatively different groups, i.e., those to set up a model, those to control the model, and those fed to the model to produce an output [although it may appear just one group from a field scientists' perspective as they are measured and observed at the same time at the field sites, except for "annual air temperature range")]. Nevertheless, lack of clear distinction between these groups appears to lead to confusions in the analysis and/or interpretations in Sections 4 and 5.

Of the variables listed as "input" in Table 1, moisture content to thawing n-factor provide the site-specific information in this model's framework, and actually determine the condition of the (under)ground at which a periglacial feature occurs. They deserve to be called "parameters" as they determine the shape and functionality of the model. To the contrary, active layer thickness is a result of action that occurred at such a place as set by the above "parameters" under a certain climate condition (i.e., thermal, in this model's case), which is the targeted output of the model. Thus, it works as the "input" or driving term of the model.

Temperature range may be called a controlling parameter to the MAAT as the two of them cannot be determined uniquely by $I_{ta}$ alone (this is also relevant to the next issue).

AC: We have encountered the terminology used in some other articles as well. Nonetheless, we accepted the referee's arguments and substituted the term "driving parameters" with "input variables" throughout the whole manuscript.

RC1: 2. Functionality of "annual air temperature range" $A_a$

When deriving MAAT from Eq (5 to 7) under the given value of $I_{ta}$, it is trivial that MAAT decreases as $A_a$ increases (leading to increase in the absolute value of $I_{fa}$, decrease in MATCM and MATTS, and increases in $L_f$), in which the value of $A_a$ directly controls the output. From this point of view, the arguments in Section 5 (namely, ll. 367–372) look off the mark. In contrast, the modelled MATCM (or other variables related to the freezing or cold season) can be used to evaluate the plausible value of $A_a$. For example, the argument shown in section 5.2 (ll. 348–350) could be reversed to discuss possible inference on the annual air temperature range that best explains the value of MATCM (or similar variables) derived from other proxies (e.g., -27 to -16.5°C in case of Central European lowland).

AC: We agree that the described relations are obvious from the equations themselves, but it may not be clear to everyone. Moreover, even though it is evident that the annual air temperature range controls the outputs, we do not want to stay on that statement, and we intend to quantify how it affects these outputs and what are its relations to other inputs. This is why we left the section (ll. 367–372) as it was.

As for the annual air temperature range, its scenarios were limited from above on the basis of maximum values indicated by other proxy-records. From this point of view, the scenarios should be plausible (or cannot be made more plausible based on these other proxies).

**RC1:** 3. Evaluations on the empirical reconstruction methods.
(This is more or less a diplomatic suggestion.) In Abstract and Introduction section, the authors state that their model is aimed to overcome the "flaws" of the empirical methods which are "far from reliable". Yet, the evaluation of the model performance did rely on the outcomes from the empirical methods (ll. 11–12, Section 5.2). Although an assertion of novelty and superiority of the new method is understandable, it doesn't appear fair. The spirit of the new model should lie in its capability to provide more verifiable reconstructions "in a replicable and subjectivity-suppressed manner" (l. 419).

**AC:** We agree, which is why we completely removed the word "flaws" and "flawed" from the text or changed the statement so that it tells that the empirical methods are problematic. Likewise, the phrase "far from reliable" was substituted with "have a/be of limited validity".

**RC1:** Minor issues/technical issues:
ll. 19–20, "Commonly,… of past environmental conditions": any reference to support the sentence?

**AC:** We added two citations (Washburn (1980) and French (2017) [see manuscript for both citations]) that support that statement.

**RC1:** l. 102: "the number of inputs" should be small.

**AC:** We changed it to "the number of inputs small".

**RC1:** l. 121: Should be Eq (5 to 7) to include boundary conditions.

**AC:** Note that the equations (6) and (7) have a solution for $-A_a / 2 \leq \text{MAAT} \leq A_a / 2$, but we target exclusively those mean annual air temperatures that are below or equal 0 °C (~permafrost conditions), and thus we solve the equations only for $-A_a / 2 < \text{MAAT} \leq 0$. This is why we think that the inclusion of the boundary conditions into the equations (6) and (7) could be more confusing than explanatory. We moved the section stating that the equations are solved for $-A_a / 2 < \text{MAAT} \leq 0$ (originally in the next paragraph) just after these equations, which should be enough to clarify this matter.

**RC1:** Table 2: It would be good to provide the number of samples (or sampling points).

**AC:** We tried to implement the numbers of samples into the Table 2, but because the numbers are mostly the same for both locations or for multiple variables and also because the table did not fit the page in that case, we decided not to include them into the table. Note, however, that the numbers of samples are mentioned in the surrounding text (now also for the palaeo-active-layer thickness).

**RC1:** l. 230, "supposed to be representative for former conditions as such": not clear. Meant something like "supposed to be unchanged from the time of cryoturbation"?

**AC:** The text was changed to "supposed to be unchanged since the cryoturbations developed."

**RC1:** Section 5.1: It should be mentioned in the preamble that this section considers the results of Section 3

(present-day application).

**AC:** We added a reference to the Section 3 and also slightly changed the text in the preamble as follows: "Generally, the model validation using the modern data (Sect. 3) showed …"

**RC1:** l. 294, l. 416: How is the "success rate" defined and evaluated?

**AC:** We removed this phrase (including the abstract) or substituted it by the word "accuracy".

**RC1:** ll. 311–314, ll. 362–365, ll. 412–415: Sentences are too long, and not clear.

**AC:** The first sentence was simplified as follows: "Surely, the Stefan equation (Eq. 2) might be improved by a number of correction factors, but these require additional inputs, such as frozen thermal conductivity, thawed and frozen volumetric heat capacity, or active-layer temperature at the start of its thawing (Kurylyk and Hayashi, 2016)."

The second sentence was simplified as follows: "Global sensitivity analysis using multiple regression suggested that the palaeo-active-layer thickness and annual air temperature range had a major impact on the modelled palaeo-air temperature characteristics at the Brno–Černovice and Nebanice site (Fig. 7). The palaeo-active-layer thickness importantly showed the highest values of the standardized regression coefficients (SRCs) especially for the annual and thawing-season air temperature attributes."

The third sentence was shortened to: "The palaeo-MAAT modelled for two sites in the Czech Republic hosting relict cryoturbation structures was between −7.0±1.9 °C and −3.2±1.5 °C and its corresponding reduction was between −16.0 °C and −11.3 °C in comparison with the 1981–2010 period, which is relatively well in line with earlier reconstructions utilizing various palaeo-archives."

**RC1:** Section 5.2: It should be mentioned in the preamble that this section considers the results of Section 4.3 (palaeo application).

**AC:** We added a reference to the Section 4.3 and also slightly changed the text in the preamble as follows: "The palaeo-MAAT modelled for two sites in the Czech Republic (Sect. 4.3) was between −7.0±1.9 °C and −3.2±1.5 °C …"

**RC1:** ll. 357–360: Additional evidence or arguments would be required to support or substantiate the claim that not the model outputs but the empirical MAAT thresholds are to be revised.

**AC:** The text was changed as follows: "This study thus raises questions about the validity of the previously suggested MAAT thresholds for cryoturbation structures (see Vandenberghe, 2013; French, 2017) and calls for their thorough revision."

**RC1:** Section 5.4: It would be suggested to modify the title, for example, "Limitations and applicability of the model".

**AC:** The section title was changed to "Model applicability to periglacial features".

**RC1:** l. 381: "However, it can also be easily adapted for seasonal-frost features": It won't be that "easily". Basically, adaptation will be a mirror image (e.g., changing the suffix $t$ to $f$), but the estimation and validation of

snow conditions (or freezing n-factor) can still be complicated.

**AC:** As the referee also states, the adaptation for freezing conditions itself would be simple, but we agree that the estimation of snow conditions via the freezing *n*-factor would be complicated. We added it into the text, which is now as follows: "Also, it could be easily adapted for seasonal-frost features, although the estimation of snow conditions would be complicated."

**RC1:** l. 383, "involving natural climate as well as active-layer thickness variations": Suggested to revise, e.g., "involving natural variations in climate as well as in active-layer thickness"?

**AC:** The sentence was changed as suggested.

**RC1:** ll. 388–390, "some periglacial features,… microstructures": "small-scale periglacial features" would suffice.

**AC:** We changed to "Indeed, some periglacial features may be produced …"

**RC1:** l. 398: What does "co-occurring periglacial features" mean? Periglacial features occurring side-by- side?

**AC:** Yes, it should mean periglacial features that occur side-by-side. We changed it to "coexisting periglacial features".

**RC1:** l. 401: "a more complete" to "an abundant"?

**AC:** We left this part of the text in its original form as it should mean that periglacial assemblages of different ages could provide a longer history of past temperatures (that is, a more complete [comprehensive] record).

**Author's response to comments of Referee #2**

**AC:** We also thank the referee #2 for the review of our manuscript.

**RC2:** l. 20: Suggestion to remove "…, or emerged at different times, …"

**AC:** We left this section in the text because permafrost-related features frequently form in extreme environments where vegetation can be sparse or absent (Washburn, 1980; French, 2017; Ballantyne, 2018 [see manuscript for all citations]). Consequently, there may be a limited amount of, for instance, biological palaeo-indicators for periods when the features formed, although these bio-indicators may be abundant for younger periods.

**RC2:** l. 30: Agreement with that the presence and distribution of periglacial features also depends on such factors as ground physical properties, hydrology, topography, or ground-surface cover: "Yes. The problem is that modeled permafrost temperature relates closer to air temperature, but the correlation between MAAT and ALT is much weaker. See paper of Wang et al., 2019: Wang, K., Overeem, I., Jafarov, E., 2019. Sensitivity Evaluation of the Kudryavtsev Permafrost Model. Science of the Total Environment. DOI: 10.1016/j.scitotenv.2020.137538"

**AC:** Given that the referee only agreed with that statement, we did not make any changes in the manuscript based on this comment.

**RC2:** l. 55: Suggestion to remove "palaeo-environmental significance as well as".

**AC:** This is largely subjective, but we left this phrase in the text because we believe that active-layer features (~permafrost features) are indicators of at least former presence of permafrost and temperatures below 0 °C, whereas seasonal-frost features frequently occur in places where seasonal freezing is still active today and they also have no uppermost temperature threshold.

**RC2:** l. 64: Comment to the statement that active-layer thawing is governed by ground-surface temperature at the surface boundary in the Stefan equation: "this is a big shortcoming too, ignores vegetative top layer and snow effects".

**AC:** Subsurface heat transfer in perhaps all analytical as well as numerical models is driven by ground-surface temperatures, which can be obtained by in situ measurements or via additional model components solving the energy exchange between the ground surface and atmosphere in various ways. This is done using so-called *n*-factors that convert between ground-surface and air thawing index in our model. Similarly, this simple but reliable *n*-factor approach was extensively utilized elsewhere mostly by analytical but also numerical models (e.g. Etzelmüller et al., 2011; Westermann et al., 2015 [see manuscript]; Obu et al., 2019). It is thus definitely not true that our model ignores surface effects.

Etzelmüller, B., Schuler, T.V., Isaksen, K., Christiansen, H.H., Farbrot, H., and Benestad, R.: Modeling the temperature evolution of Svalbard permafrost during the 20th and 21st century. The Cryosphere, 5, 67–79, https://doi.org/10.5194/tc-5-67-2011, 2011.
Obu, J., Westermann, S., Bartsch, A., Berdnikov, N., Christiansen, H.H., Dashtseren, A., Delaloye, R., Elberling, B., Etzelmüller, B., Kholodov, A., Khomutov, A., Kääb, A., Leibman, M.O., Lewkowicz, A.G., Panda, S.K., Romanovsky, V.,Way, R.G.,Westergaard-Nielsen, A.,Wu, T., Yamkhin, J., and Zou, D.: Northern Hemisphere permafrost map based on TTOP modelling for 2000–2016 at 1 km$^2$ scale. Earth-Science Reviews, 193, 299–316, https://doi.org/10.1016/j.earscirev.2019.04.023, 2019.

**RC2:** l. 240: Comment to the statement that treeless landscapes dominated in South Moravia and West Bohemia at the time when cryoturbations formed: "you may want to emphasize this approach a bit more in the discussion. Perhaps local pollen records can always be part of the puzzle and give better idea of snow thickness and vegetation effects?"

**AC:** We agree that pollen records could give a better idea of vegetation type and coverage, which might be useful for thawing *n*-factor estimation. However, the analysis should ideally be done on the same profile where cryoturbations occur, which is a good suggestion for follow-up detailed studies, but we did not do it in this study, which should chiefly describe and evaluate the model. Instead, we relied on a general lack of organic remains within the cryoturbated horizons (this argument was added into the revised manuscript) and regional pollen-based reconstructions (Kuneš et al., 2008 [see manuscript]).

**RC2:** l. 403–405: Suggestion to remove "Nonetheless, this shortcoming is also increasingly being suppressed by improved dating methods that bring more reliable periglacial chronologies (e.g., Andrieux et al., 2018; Nyland et al., 2020; Engel et al., in print)."

**AC:** We left this sentence in the text because it can show readers thinking that periglacial features cannot be dated that it is becoming more reliable than before. Indeed, dating now provides meaningful ages that can be combined with derived palaeo-air temperatures.